# Exploring the principles behind antibiotics with limited resistance

Elvin Maharramov [1,2,8], Márton Simon Czikkely[1,3,4,8], Petra Szili[1], Zoltán Farkas [1], Gábor Grézal [1,5], Lejla Daruka[1], Eszter Kurkó[1], Léna Mészáros[6], Andreea Daraba[1], Terézia Kovács[1,7], Bence Bognár[1,2,7], Szilvia Juhász[1,6], Balázs Papp [1,5], Viktória Lázár[1,7] & Csaba Pál [1] ✉

Antibiotics that target multiple cellular functions are anticipated to be less prone to bacterial resistance. Here we hypothesize that while dual targeting is crucial, it is not sufficient in preventing resistance. Only those antibiotics that simultaneously target membrane integrity and block another cellular pathway display reduced resistance development. To test the hypothesis, we focus on three antibiotic candidates, POL7306, Tridecaptin M152-P3 and SCH79797, all of which fulfill the above criteria. Here we show that resistance evolution against these antibiotics is limited in ESKAPE pathogens, including *Escherichia coli*, *Klebsiella pneumoniae*, *Acinetobacter baumannii* and *Pseudomonas aeruginosa*, while dual-target topoisomerase antibiotics are prone to resistance. We discover several mechanisms restricting resistance. First, de novo mutations result in only a limited elevation in resistance, including those affecting the molecular targets and efflux pumps. Second, resistance is inaccessible through gene amplification. Third, functional metagenomics reveal that mobile resistance genes are rare in human gut, soil and clinical microbiomes. Finally, we detect rapid eradication of bacterial populations upon toxic exposure to membrane targeting antibiotics. We conclude that resistance mechanisms commonly found in natural bacterial pathogens provide only limited protection to these antibiotics. Our work provides guidelines for the future development of antibiotics.

Nearly 50 years have passed since the golden age of antibiotic discovery (1945–1975) ended, and yet we continue to struggle to develop antibiotics with novel modes of action. Even though valiant attempts have been made to leverage a variety of antibiotic discovery platform approaches (genomics, bioinformatics, systems biology, and postgenomic techniques), progress remains incremental at best[1]. This slow progress, coupled with the rapid spread of multi-drug resistant bacteria, has led numerous pharmaceutical companies to abort their antibiotic research initiatives, thereby making the commercial success of prospective antimicrobial drugs largely unpredictable[2,3]. Hence, it is crucial to identify the key factors shaping the risk of developing resistance and incorporate this knowledge into the antibiotic development process.

[1]Synthetic and Systems Biology Unit, Institute of Biochemistry, HUN-REN Biological Research Centre Szeged, Szeged, Hungary. [2]Doctoral School of Biology, University of Szeged, Szeged, Hungary. [3]Doctoral School of Multidisciplinary Medical Sciences, University of Szeged, Szeged, Hungary. [4]Department of Forensic Medicine, Albert-Szent-Györgyi Medical School, University of Szeged, Szeged, Hungary. [5]HCEMM-BRC Metabolic Systems Biology Group, Szeged, Hungary. [6]Hungarian Centre of Excellence for Molecular Medicine, Cancer Microbiome Core Group, Budapesti út 9, Szeged, Hungary. [7]HCEMM-BRC Pharmacodynamic Drug Interaction Research Group, Szeged, Hungary. [8]These authors contributed equally: Elvin Maharramov, Márton Simon Czikkely. ✉e-mail: cpal@brc.hu

In response to the need for effective therapeutic options, two promising avenues have emerged: i) multi-targeting antibiotics, which include single compounds capable of inhibiting multiple bacterial targets, and ii) antibiotics targeting immutable elements of bacterial physiology[4,5]. The first case comprises single compounds that inhibit multiple non-overlapping molecular targets. The rationale is that resistance development would require the simultaneous occurrence of multiple mutations. However, this approach has its challenges, as it assumes that single mutations offer minimal or no advantage to bacteria. These single mutations are often found in natural bacterial populations, providing significant resistance against other, clinically employed antimicrobial agents. Prolonged antibiotic use might inadvertently promote the spread of mutations acting as stepping stones toward resistance to dual-target antibiotics still in development[6]. Furthermore, bacteria can gain resistance through various mechanisms unrelated to target mutations, including enzymatic breakdown, alterations in membrane permeability, and variations in the activity or specificity of bacterial efflux pumps[7].

The second solution is to develop antibiotics with non-mutable targets, such as non-protein based cellular structures that are not directly DNA-encoded. These antibiotics may target one function only, but resistance seriously compromises bacterial survival or virulence. Compounds that attack the integrity of the cell envelope could be pivotal here, given the distinctive properties of bacterial cell surfaces which are often not directly protein-encoded, making them less susceptible to mutation-induced alterations[3,8]. For example, it has been suggested that Lipid II, an intermediate in peptidoglycan biosynthesis presents a new opportunity for drugs with limited susceptibility to resistance[9,10].

Based on these prior considerations[3–5,8,11], we hypothesize that to limit resistance development, new antibiotics should have dual-modes of action and specifically target membrane integrity. In contrast, antibiotics with two protein-encoded intracellular targets remain susceptible to resistance development.

Specifically, we focus on three antibacterial agents that permeabilize the bacterial outer membrane and putatively have dual modes of action (dual-target (DT) permeabilizers). POL7306 attaches to lipopolysaccharides (LPS) and the crucial outer membrane protein BamA. BamA plays a key role in the folding and insertion of β-barrel proteins into the outer membrane of Gram-negative bacteria[12]. Tridecaptin M152-P3 binds primarily to lipid II thus blocking ATP synthesis by dissipating the proton motive force and additionally binds to LPS[13–15]. SCH79797 damages the bacterial membrane by directly activating the bacterial channel MscL, thereby causing membrane permeabilization[16–19], while also hindering folate biosynthesis. The large conductance mechanosensitive channel in bacteria, MscL, typically acts as an osmotic emergency valve that opens during hypoosmotic shock, influencing membrane permeabilization. Although the exact modes of action of these antibiotic candidates differ, all of them permeabilize the bacterial outer membrane (Supplementary Figs. 1, 2).

As controls, we study three main groups of modes of action, all of which directly act on Gram-negative bacteria, representing various primary antibiotic classes. The first group is composed of gepotidacin, delafloxacin, zoliflodacin and moxifloxacin, all of which target two homologous intracellular sites (DT non-permeabilizers)[6,20–23]. Although their chemical structures and their precise binding sites vary, these antibiotics selectively inhibit bacterial DNA gyrase and topoisomerase IV complexes. The extent to which these new antibiotics are prone to resistance is a subject of intense debate[6,23–25]. The second group includes peptide-based antibiotics like polymyxin B and SPR206, which act by a general membrane lysis mechanism, similarly to numerous cationic antimicrobial peptides (single-target (ST) permeabilizers)[26,27]. The third group includes other mono-targeting antibiotics each with its unique antibacterial mechanism that is independent of membrane lysis (ST non-permeabilizers; for details and abbreviations see Fig. 1 and Table 1).

In an earlier work, we studied the susceptibility of these antibiotics to resistance development in one multi-drug resistant and one sensitive strain each of *Escherichia coli*, *Klebsiella pneumoniae*, *Acinetobacter baumannii* and *Pseudomonas aeruginosa*[28]. These Gram-negative species are among the critical-priority pathogens according to the World Health Organization (WHO)[29]. Using a conventional method for spontaneous resistance frequency analysis (FoR assay) at varying antibiotic concentrations, we sought to understand de novo resistance emergence. We also initiated adaptive laboratory evolution (ALE) to see how antibiotic resistance in the populations might increase over a more extended, yet fixed timeframe (roughly 120 generations). Elevated resistance was observed in ~40% (for FoR assay) and ~91% (for ALE) of the antibiotic-strain combinations we studied[28]. We next extended our focus to include horizontally transferred resistance mechanisms. We investigated the prevalence of horizontally transferred antibiotic resistance genes in diverse resistomes, encompassing both environmental and clinical sources. Our analysis encompassed metagenomic libraries derived from three distinct sources: i) river sediment and soil samples from seven antibiotic-contaminated industrial sites near antibiotic manufacturing plants in India, representing the anthropogenic soil microbiome; ii) fecal samples from ten Europeans with no antibiotic consumption for at least one year before collection, representing the gut microbiome; and iii) a composite sample from 68 multi-drug resistant bacteria, either isolated in medical settings or acquired from bacterial strain collections, representing the clinical microbiome. Utilizing established functional metagenomic techniques[30], we identified 690 unique DNA segments capable of significantly enhancing resistance to one of each antibiotic[28].

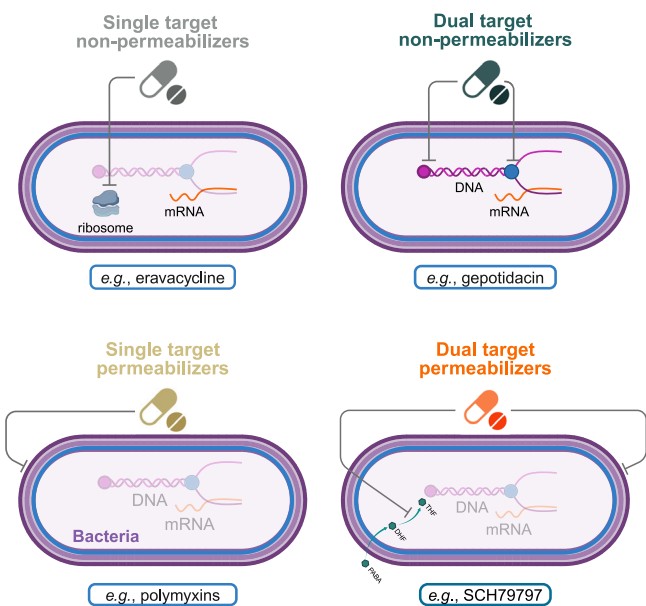

**Fig. 1 | The four main modes of action groups used in our study.** The figure shows examples of the mechanisms of action for each group. Single target non-permeabilizers (ST non-permeabilizers) target a single cellular component without any effect on membrane integrity, exemplified by eravacycline, a protein synthesis inhibitor. Dual target non-permeabilizers (DT non-permeabilizers) have two main intracellular targets, without effect on membrane integrity, exemplified by gepotidacin, which targets DNA gyrase and topoisomerase IV. Single target permeabilizers (ST permeabilizers), such as polymyxins, disrupt the bacterial cell membrane. Dual target permeabilizers (DT permeabilizers) target membrane integrity and an additional cellular pathway concurrently, exemplified by SCH79797, which disrupts both the cell membrane and folate synthesis. Created in BioRender. Czikkely, M. (2025) https://BioRender.com/k95g583.

**Table 1 | The antibiotics employed in this study**

| Antibiotic | Abbreviation | Antibiotic Class | Mode of action group | Date of approval/clinical phase |
|---|---|---|---|---|
| Omadacycline | OMA | Tetracycline | Single target non-permeabilizer | 2018 |
| Eravacycline | ERA | Tetracycline | Single target non-permeabilizer | 2018 |
| Doxycycline | DOX | Tetracycline | Single target non-permeabilizer | 1967 |
| Apramycin | APR | Aminoglycoside | Single target non-permeabilizer | Phase 1 |
| Gentamicin | GEN | Aminoglycoside | Single target non-permeabilizer | 1964 |
| Sulopenem | SUO | Carbapenem | Single target non-permeabilizer | Phase 3 |
| Meropenem | MER | Carbapenem | Single target non-permeabilizer | 1996 |
| Delafloxacin | DEL | Topoisomerase inhibitor | Dual target non-permeabilizer | 2017 |
| Gepotidacin | GEP | Topoisomerase inhibitor | Dual target non-permeabilizer | Phase 3 |
| Zoliflodacin | ZOL | Topoisomerase inhibitor | Dual target non-permeabilizer | Phase 3 |
| Moxifloxacin | MOX | Topoisomerase inhibitor | Dual target non-permeabilizer | 1999 |
| Tridecaptin M152-P3 | TRD* | Targeting membrane integrity and an additional cellular pathway | Dual target permeabilizer | pre-clinical |
| POL7306 | POL* | Targeting membrane integrity and an additional cellular pathway | Dual target permeabilizer | pre-clinical |
| SCH79797 | SCH* | Targeting membrane integrity and an additional cellular pathway | Dual target permeabilizer | pre-clinical |
| SPR206 | SPR | Polymyxin | Single target permeabilizer | Phase 2 |
| Polymyxin B | PMB | Polymyxin | Single target permeabilizer | 1964 |

We divided the mode of action (MoA) groups into single target non-permeabilizers, dual target non-permeabilizers, single target permeabilizers, and dual target permeabilizers. Tridecaptin M152-P3, POL7306, and SCH79797, all denoted with an asterisk, are in the pre-clinical phase and are identified as dual target permeabilizers.

Here we show, by integrating these data with additional experimental screens, that dual-target permeabilizers, including POL7306, tridecaptin M152-P3, and SCH79797 are less susceptible to resistance in Gram-negative bacterial pathogens than any other antibiotic groups. In addition, we demonstrate that mobile resistance genes for these compounds are rare in human-associated microbiomes. In-depth functional analyzes revealed that typical resistance pathways are not readily accessible, and the antibiotics killed bacterial populations very effectively. Finally, we provide suggestions on how future dual-target permeabilizer antibiotic candidates could be further developed to remain effective in the long-term.

## Results

### Bacterial resistance to dual-target permeabilizer antibiotics is limited

The susceptibility of antibiotics to resistance development was evaluated in two strains of each bacterial species: a multidrug-resistant strain and an antibiotic-sensitive strain of *Escherichia coli*, *Klebsiella pneumoniae*, *Acinetobacter baumannii*, and *Pseudomonas aeruginosa*. To explore first-step resistance, we previously used a standard protocol for spontaneous frequency-of-resistance analysis (FoR assay) at multiple concentrations of each antibiotic[10,28,31–33]. Antibiotic-strain combinations exhibiting decreased initial susceptibility (*e.g.*, > 4 µg/mL) were excluded from the experiments (see Supplementary Data 1). For all other combinations, approximately $10^{10}$ bacterial cells were exposed to one of each antibiotic on agar plates for two days at concentrations where the given strain is susceptible. We examined resistance levels by contrasting the minimum inhibitory concentration (MICs) of the mutated strains to the original wild-type bacterial strain (*i.e.*, relative MIC).

In all bacterial species, resistance typically developed quickly in the three control groups, including ST permeabilizer, DT non-permeabilizer and ST non-permeabilizer antibiotics. By contrast, there was a low probability of resistance development against DT permeabilizers. In particular, the average increment in resistance level was less than fourfold in populations exposed to POL7306, tridecaptin M152-P3, and SCH79797 stresses (Fig. 2a).

In stark contrast, resistance levels increased by more than 128-fold for polymyxin B in *A. baumannii* multidrug-resistant (MDR) strain and SPR206 in *E. coli* and *K. pneumoniae* sensitive strains (Supplementary Fig. 3). Notably, no resistant mutants emerged during this assay for polymyxin B in *A. baumannii* and *K. pneumoniae* sensitive strains, nor in the multidrug-resistant *E. coli* strain. However, the underlying FoR assay cannot detect very rare mutations and combinations thereof, and hence they may underestimate bacterial potential for resistance[32]. Therefore, it is essential to demonstrate how long-term exposure to these antibiotics affects resistance evolution.

Using the same set of bacterial strains, we initiated adaptive laboratory evolution with the aim to maximize the level of antibiotic resistance in the populations achieved during a longer, but fixed time period[28]. As expected, resistance levels were generally much higher than those observed in FoR assay. Reassuringly, the level of resistance against DT permeabilizers was significantly lower in comparison to all other antibiotic groups (Fig. 2b). For instance, after 60 days of evolution, resistance levels increased by a maximum of fourfold in *A. baumannii* and in the *E. coli* sensitive strain exposed to SCH79797. By contrast, polymyxin B-resistant lineages displayed over 1024-fold increments in resistance levels.

Of note, when analyzing all antibiotic pairs on a strain-specific basis, DT permeabilizers consistently exhibited significantly lower median resistance levels than ST permeabilizers in 69% of the cases (Supplementary Fig. 4). Importantly, no instances were detected where an ST permeabilizer resulted in a lower relative MIC compared to any DT permeabilizer. Furthermore, the capacity to develop resistance against POL7306 and SCH79797 displayed significant heterogeneity across bacterial species (Fig. 3b; Kruskal-Wallis rank-sum test $p = 0.000011$ and $p = 0.0027$, respectively). For example, *A. baumannii* showed elevated resistance to POL7306 only, whereas *K. pneumoniae* exhibited increased resistance to both POL7306 and SCH79797 (Fig. 3b). It is important to note that due to *P. aeruginosa*'s decreased initial susceptibility (*e.g.*, > 4 µg/mL) to other DT permeabilizers, only POL7306 was analyzed in this species. Notably, resistance level reached after adaptive laboratory evolution was significantly lower compared to ST permeabilizers (Supplementary Fig. 4, Dunn's

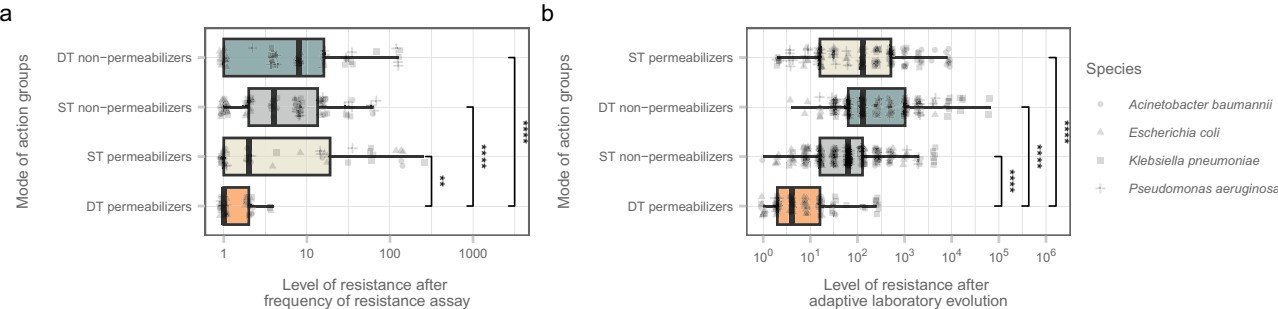

**Fig. 2 | Limited resistance to DT permeabilizers.** Resistance levels (**a**), after frequency of resistance assay and (**b**), after adaptive laboratory evolution were assessed by using relative Minimum Inhibitory Concentration (MIC) values calculated by dividing the MIC of the evolved line by that of the corresponding ancestor. Each data point represents an independent evolved line, with species differentiation denoted by shape. Significant variation in resistance levels was observed across different modes of action groups (Kruskal-Wallis test, **a**: $n = 245$, chi-squared = 28.312, df = 3, $p = 3.12 \times 10^{-6}$; **b**: $n = 728$, chi-squared = 205.014, df = 3,

$p = 3.48 \times 10^{-4}$). Next, significant difference was observed in the relative MIC of DT permeabilizers compared to other groups (two-sided Dunn's post-hoc test with Benjamini-Hochberg correction, ** / **** indicates $p < 0.01 / 0.0001$, respectively). The boxplots show the median, first and third quartiles, with whiskers showing the 5th and 95th percentiles. Non-significant $p$-values were excluded from the figure. Source data are provided as a Source Data file. For mode of action groups, refer to Table 1.

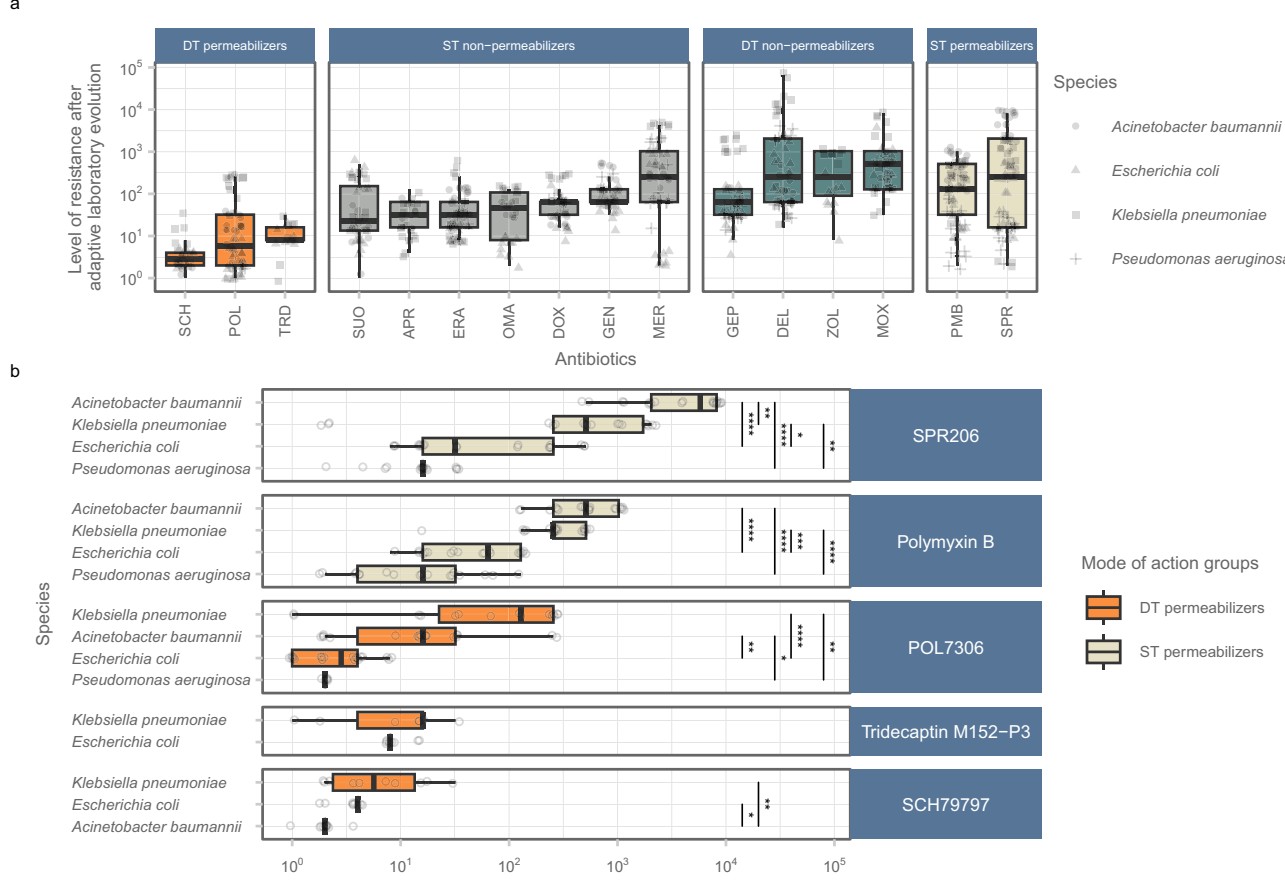

**Fig. 3 | Comparative antibiotic resistance profiles after adaptive laboratory evolution. a** Antibiotic-specific differences in resistance levels. The figure shows the resistance levels of different antibiotics, categorized by their action as DT permeabilizers, ST non-permeabilizers, DT non-permeabilizers, and ST permeabilizers (panels). The data suggest a significant variation in resistance development across all antibiotics ($n = 728$, Kruskal-Wallis chi-squared = 281.03, df = 15, $p = 4.83 \times 10^{-51}$). **b** Species-specific differences in resistance levels. Species-specific resistance patterns were evident, particularly with POL7306 and SCH79797 ($n = 60$, Kruskal-Wallis chi-squared = 25.6, df = 3, $p = 1.14 \times 10^{-5}$ and $n = 30$, chi-squared = 11.86, df = 2, $p = 0.0027$, respectively). *A. baumannii* and *K. pneumoniae* showed heightened resistance to POL7306, whereas *K. pneumoniae* exhibited increased resistance to SCH79797.

Statistical analysis was performed using two-sided Dunn's post-hoc test with Benjamini-Hochberg correction for multiple comparisons (**** / *** / ** / * indicates $p < 0.0001/ 0.001/ 0.01/ 0.05$). The level of resistance was assessed using relative MIC values, calculated by dividing the MIC of the evolved line by that of the corresponding ancestor. Antibiotic-strain combinations with reduced initial susceptibility (MIC > 4 µg/mL) were excluded from evolution experiments. Each data point represents a unique adaptive laboratory-evolved line. Species distinction is indicated by shape, while mode of action group by color. The boxplots show the median, first and third quartiles, with whiskers showing the 5th and 95th percentiles. For detailed information on antibiotic and bacterial species, see Supplementary Data 1. Source data are provided as a Source Data file. For mode of action groups, refer to Table 1.

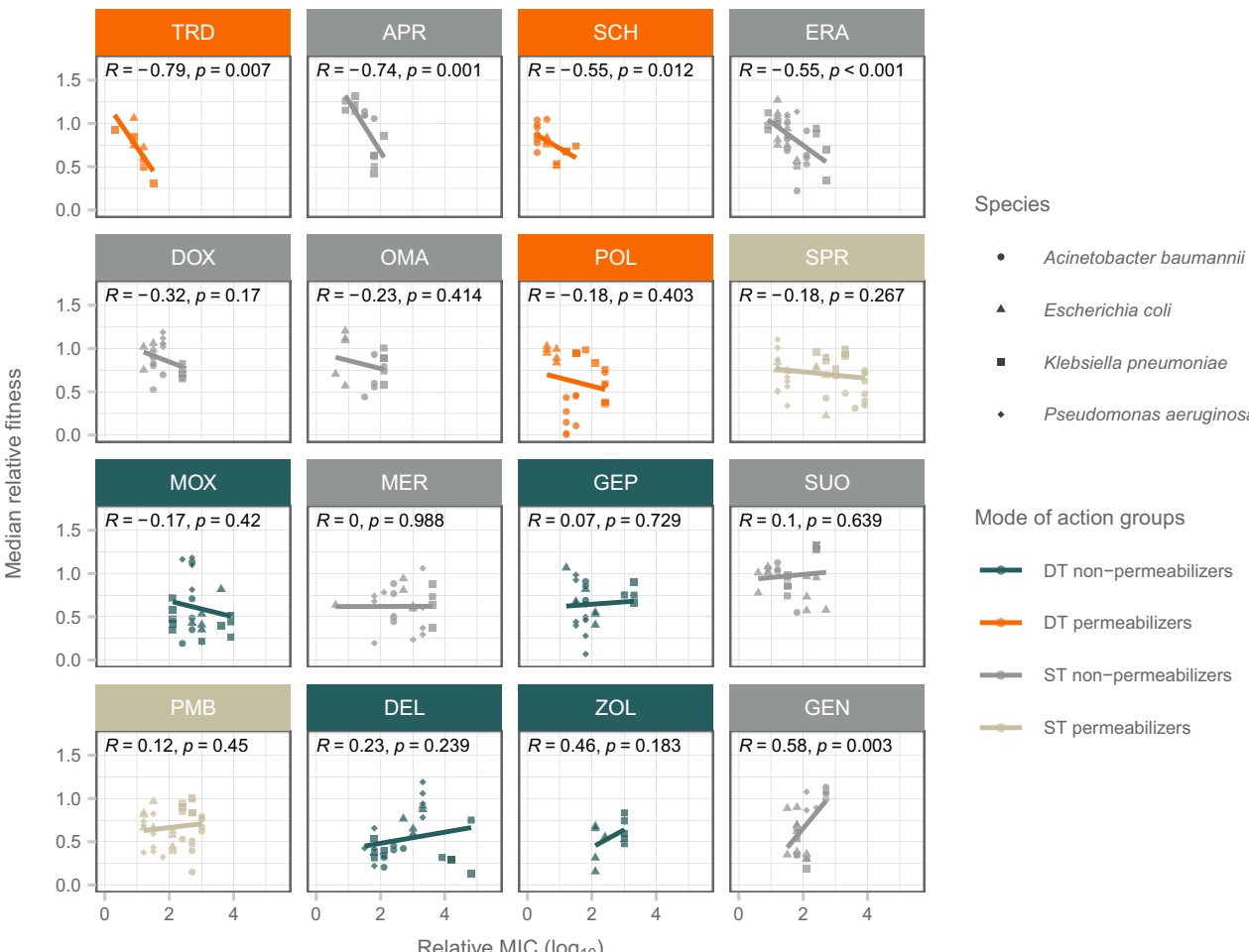

**Fig. 4 | Correlation between bacterial resistance level and bacterial fitness.** The scatterplot shows the median relative fitness of each bacterial line plotted against the $\log_{10}$-transformed relative minimum inhibitory concentration (MIC). Relative bacterial fitness was determined by comparing the area under the growth curve of antibiotic-adapted lines ($n = 385$) to their ancestral strains in an antibiotic-free environment (see "Methods"). Each point represents an evolved bacterial line, with colors distinguishing lines adapted to different modes of action groups.

A significant negative correlation between relative fitness and resistance level (relative MIC) was observed in lines adapted to tridecaptin M152-P3 and SCH79797 (both DT permeabilizers) with Pearson correlation coefficients of $R = -0.79$ and $R = -0.55$, respectively (two-sided significance test $p = 0.007 / 0.012$), and in lines adapted to apramycin sulfate and eravacycline (ST non-permeabilizers) with Pearson correlation coefficients of $R = -0.74$ and $R = -0.55$, respectively (two-sided significance test $p = 0.001 / 0.0008$). Source data are provided as a Source Data file.

post-hoc test with Benjamini-Hochberg correction, $p < 0.01$). However, further studies, including a broader range of DT permeabilizers, are needed to validate this result.

In natural settings, the rise of resistance is frequently constrained by its effects on bacterial growth and viability[34]. Here we studied bacterial fitness by measuring in vitro growth of 385 antibiotic-adapted lines from adaptive laboratory evolution displaying elevated resistance and their corresponding ancestors in an antibiotic-free environment (see "Methods"). If the evolution of resistance were limited by its harmful impact on bacterial growth, one would anticipate a negative correlation between fitness and resistance levels[34]. This relationship was observed for two DT permeabilizers, SCH79797 and tridecaptin M152-P3 (Fig. 4; Pearson correlation, $R = -0.79$ and $R = -0.55$, respectively; $p < 0.05$), but also for two ST non-permeabilizers, apramycin sulfate and eravacycline (Fig. 4; Pearson correlation, $R = -0.74$ and $R = -0.55$, respectively; $p < 0.05$). Therefore, the lower evolved resistance levels to SCH79797 and tridecaptin M152-P3 may be partly linked to the fitness cost of resistance mutations, but further data are needed to resolve this issue.

### Resistance mechanisms to dual-target permeabilizers

We analyzed the non-synonymous mutations in protein coding genes that had accumulated during laboratory evolution and found that the mutated genes in lines adapted to DT permeabilizers were largely distinct from those in other antibiotic-adapted lines (Supplementary Data 2, Supplementary Fig. 5). We focused on the resistance mechanisms of DT permeabilizers and ST permeabilizers (Fig. 5a), as these antibiotics commonly disintegrate the bacterial membrane, and therefore the resistance mechanisms may overlap.

Surprisingly, we observed that the majority of the mutated genes (80%) were antibiotic-specific, with none of the identified mutations shared across all five antibiotics. POL7306 exhibited higher overlap in the set of mutated genes associated with ST permeabilizers (8.12%) compared to tridecaptin M152-P3 (2.5%) and SCH79797 (0.6%), as shown in Fig. 5a. This observation aligns with the fact that POL7306, a peptide-based antibiotic, shares structural and functional similarity with polymyxin B[12].

Genes involved in cell envelope biogenesis and regulation were commonly mutated in lines adapted to SPR206, polymyxin B, POL7306, and tridecaptin M152-P3 (Fig. 5b). For example, the BasS-BasR two-component sensor system and the outer membrane phospholipase PldA was mutated repeatedly in response to exposure to Polymyxin B, SPR206 and POL7306. Mutations in members of the PhoPQ gene family (*phoQ* and *qseC*)[35] and in the *lpxA* gene accumulated exclusively in lines adapted to ST permeabilizers. These genes

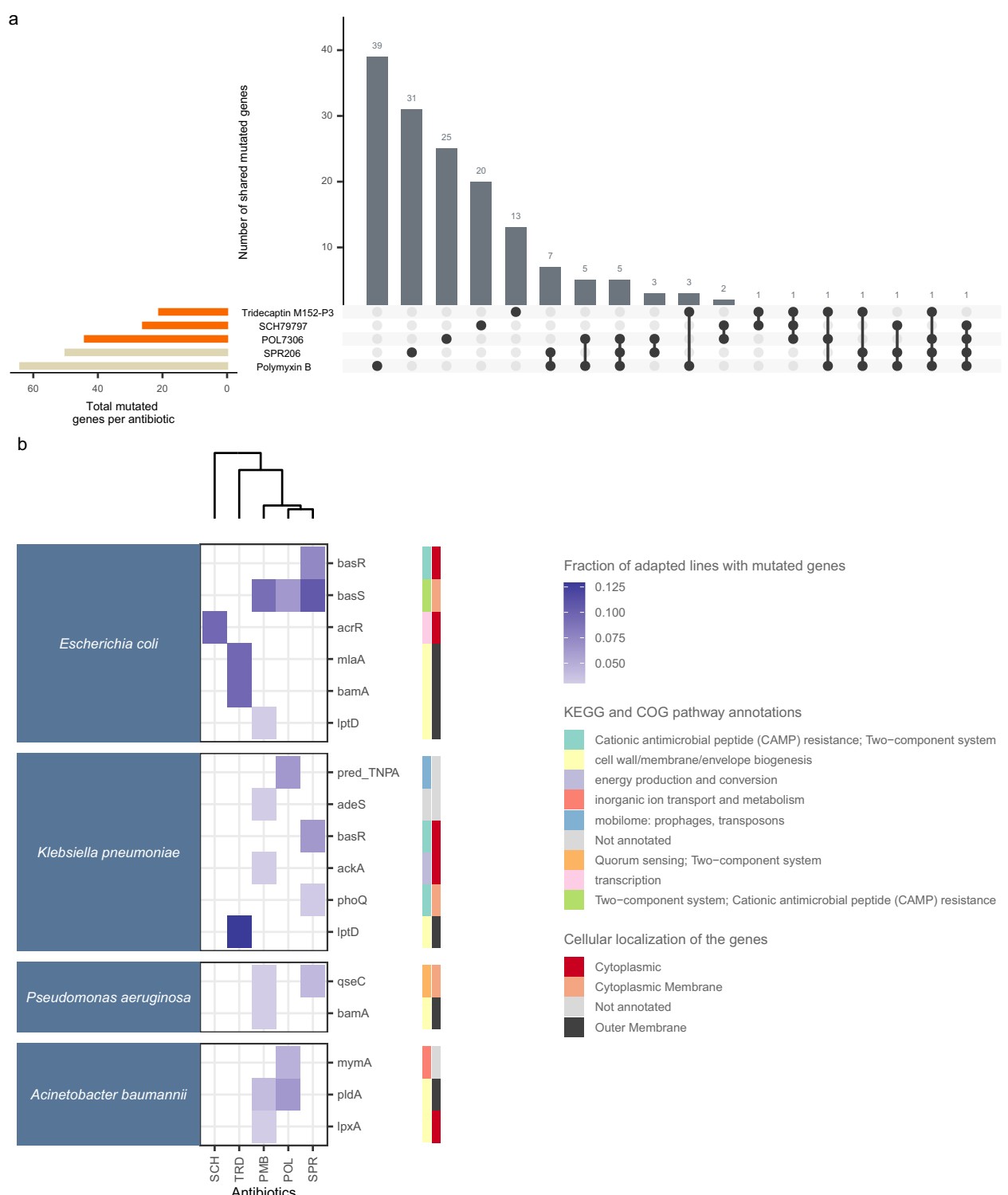

shape the assembly and structural alteration of LPS and lipid A biosynthesis[36]. Tridecaptin M152-P3 adapted lines also displayed mutations in genes involved in LPS biosynthesis/regulation and phospholipid trafficking (*lptD* and *mlaA*)[37–41].

By contrast, SCH79797 adapted lines carried mutations in the *acrR* gene, which encodes a transcription factor controlling the AcrAB-TolC multidrug efflux system[42,43] (Fig. 5b). As a small molecule with an intracellular target, it is reasonable to assume that SCH79797 is a potential subject of this efflux system, providing an unspecific, relatively low resistance level to this antibiotic. To probe this, we used

isogenic strains of *E. coli* with altered activities of the AcrAB-TolC efflux system. We found that changing efflux pump activity reduced bacterial susceptibility to a range of antibiotics, including DT non-permeabilizers and SCH79797. However, it did not affect sensitivity to POL7306, tridecaptin M152-P3, or to polymyxin B (Fig. 6).

To further investigate potential overlaps in resistance mechanisms, we examined the extent of cross-resistance between polymyxin B and other antibiotics that target the bacterial membrane. The analysis was performed on a selected set of laboratory evolved lines ($n = 12$), all of which had developed high levels of resistance to polymyxin B. We

**Fig. 5 | Analysis of mutated genes found in ST and DT permeabilizer evolved lines. a** Overlap of mutated genes in response to ST (polymyxin B, SPR206) and DT permeabilizers (SCH79797, tridecaptin M152-P3, POL7306). The plot shows the sets of mutated genes identified for each antibiotic and their intersections. The vertical bars represent the number of mutated genes shared among the indicated antibiotics, as shown by the connected dots. A single dot indicates that the mutated genes are specific to a single antibiotic and are not shared with any others. Horizontal bars indicate the total number of mutated genes per antibiotic. The majority of the mutated genes are antibiotic specific (80%, $n = 128$), and none of the identified genes were shared among all five antibiotics. **b** Composite heatmap on the molecular functions and cellular localization of the fraction of frequently mutated genes among permeabilizer antibiotics. The heatmap shows the fraction of mutated genes across all species after laboratory evolution in response to ST and DT permeabilizer antibiotics (left panel). Genes are categorized based on their cellular localization using PSORT (Protein Subcellular Localization Prediction Tool). They are further classified according to their involvement in specific metabolic or resistance pathways using the KEGG (Kyoto Encyclopedia of Genes and Genomes) and COG (Clusters of Orthologous Genes) databases. These classifications are displayed in the additional panels on the right. Each row corresponds to a unique mutated gene, with color intensity indicating the fraction of adapted lines where a specific gene was mutated. This value is calculated by dividing the number of adapted lines where the gene was mutated by the total number of adapted lines sequenced for that antibiotic. Only genes that were mutated in at least three different adapted lines are displayed. The dendrogram represents the hierarchical clustering of antibiotics based on the mutational profile similarity of genes seen in this figure. The clustering was performed using the average linkage method, grouping antibiotics with similar mutated gene profiles together. For a detailed list of genes, refer to Supplementary Data 2. Source data are provided as a Source Data file. For antibiotic abbreviations, see Table 1.

measured the susceptibilities of these lines, along with the corresponding ancestor to ST permeabilizers (SPR206 and colistin) and DT permeabilizers (SCH79797, tridecaptin M152-P3, and POL7306). We found that adaptation to polymyxin B resulted in cross-resistance to SPR206 and colistin, while susceptibility to SCH79797 and tridecaptin M152-P3 remained unchanged (Fig. 7). Among the polymyxin B resistant lines, a subset ($n = 7$) exhibited reduced susceptibility to POL7306, although these changes were less pronounced compared to those observed with SPR206 and colistin (Supplementary Fig. 6) and were specific to *K. pneumoniae* and *A. baumannii* only (Fig. 7).

Altogether, these analyzes indicate that the resistance mechanisms to dual-target permeabilizers currently in development display limited overlap with each other and to other peptide-based antibiotics that mainly target the bacterial outer membrane (polymyxin B and SPR206).

## Resistance to dual-target permeabilizers by gene amplification is limited

In nature, gene amplification is a powerful mechanism that enhances antibiotic resistance by increasing the dosage of resistance-conferring proteins, and thereby allowing for reversible adaptation to changing environments[44]. We harnessed the ASKA library[45], which entails every *E. coli* ORF cloned into an expression vector, to test resistance against all compounds using standard protocols[46]. In brief, this library was integrated into *E. coli*, and the populations were challenged with increasing levels of antibacterial stresses in a drug-diffusion assay[47]. The integrated ASKA library generated an up to 8-fold increment in antibiotic resistance level, suggesting that gene amplification can have a significant contribution to resistance.

Sequence analysis identified 67 genes which, upon elevated expression, reduced susceptibility to multiple antibiotics, many of which are involved in the transcriptional regulation of other genes. Beyond the universally recognized multi-drug resistance genes like *marA* and *soxS*[48–51], we identified several other genes with distinct molecular functions, including stringent response genes (*ytfK* and *dksA*)[52,53]. However, the integrated ASKA library did not provide any notable resistance increment against DT and ST membrane permeabilizers (Fig. 8a), and hence no gene hits were detected. This suggests that artificially amplified genes do not provide resistance to these antibiotics. However, overexpression of genes or combinations thereof may enhance bacterial growth rate under sublethal antibiotic concentrations.

## Mobile resistance genes to dual-target permeabilizers are rare in the environment

Next, we investigated the prevalence of mobile resistance genes that provide resistance to the antibiotics investigated here. Our study involved analyzing metagenomic libraries from soil, gut and clinical microbiomes, respectively (see introduction and Supplementary Data 4 for details). Using advanced functional metagenomic techniques[30], we

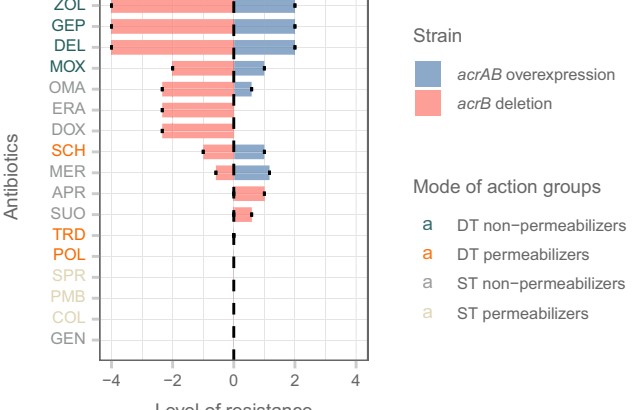

**Fig. 6 | The impact of AcrAB-TolC activity on antibiotic susceptibility.** *E. coli* strains with AcrAB-TolC depletion (BW25113-Δ*acrB*) and overexpression (BW25113:pUCacrAB) were treated with various antibiotics, and the level of resistance was determined as minimum inhibitory concentration (MIC) relative to a wild-type strain, presented on a logarithmic scale ($\log_2$). Each bar represents the mean resistance level (susceptibility, *i.e.*, level of resistance < 0; and resistance, *i.e.*, level of resistance > 0) to antibiotics induced by acrAB overexpression (blue bars; *E. coli* with a multicopy plasmid of the acrAB genes (pUCacrAB)) and *acrB* deletion (red bars; mutant *E. coli* with *acrB* gene deletion (Δ*acrB*)) compared to the wild-type. Data are presented as mean values of 3 biological replicates ± SD. The figure shows the differential impact of AcrAB-TolC expression on susceptibility to a range of antibiotics, with DT and ST permeabilizer antibiotics remaining unaffected, except for SCH79797 (SCH). Detailed information on antibiotic and bacterial strain abbreviations is provided in Supplementary Data 1. Source data are provided as a Source Data file. For information on mode of action groups, refer to Table 1.

earlier found 551 DNA segments that boosted resistance to the antibiotics tested here[28]. In total, 539 non-redundant open reading frames (ORFs) were detected, many of which were present in multiple DNA fragments.

We found that the number of resistance-conferring segments was significantly lower for DT permeabilizers compared to ST permeabilizers and DT non-permeabilizers (Fig. 8b, Dunn's post-hoc test with Benjamini-Hochberg correction, $p < 0.05$). Notably, when focusing exclusively on antibiotics currently in development, DT permeabilizers again showed a significantly reduced number of resistance-conferring segments compared to the others (Supplementary Fig. 7; Student's *t*-test, $p < 0.01$). The study of tridecaptin M152-P3 revealed an important outcome: no resistance-conferring DNA contigs were detected following selection, underscoring its potential robustness against resistance development (Fig. 8b). Additionally, SCH79797 showed the second-lowest count of DNA contigs (n = 17). These findings highlight the unique potential of DT permeabilizers to mitigate LPS modification-based resistance in natural environments. In addition, we

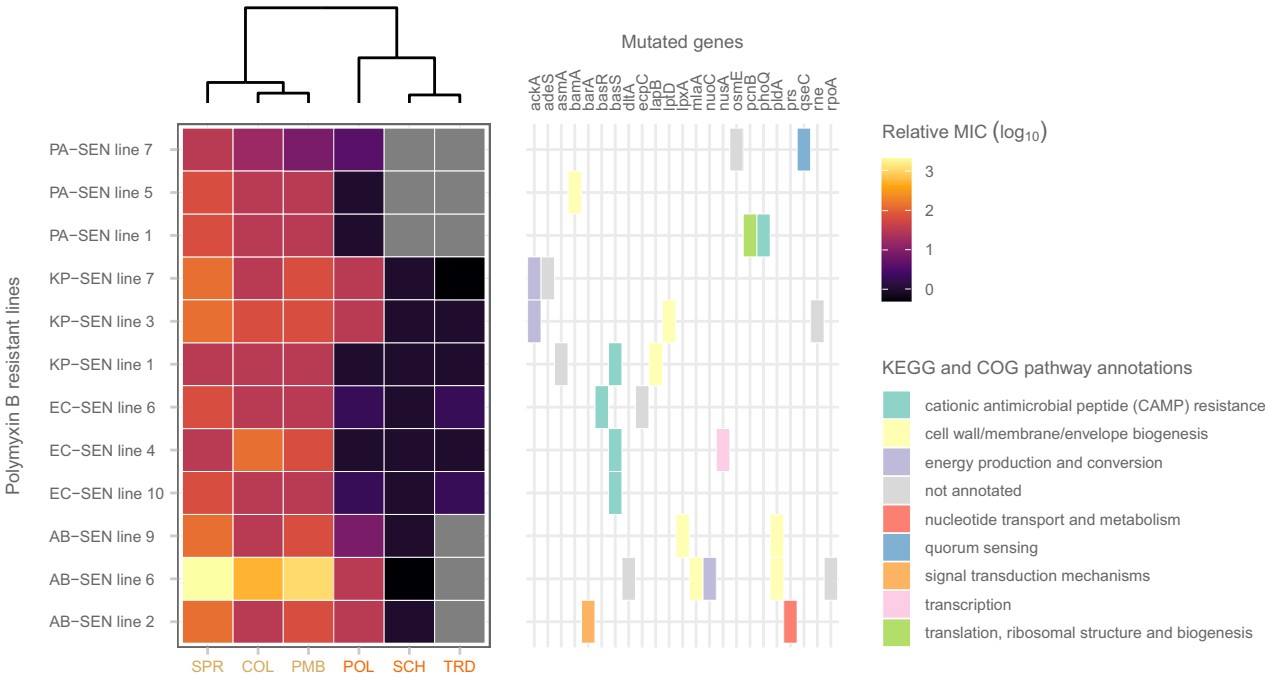

**Fig. 7 | Cross-resistance of polymyxin B resistant lines to ST and DT permeabilizer antibiotics.** The heatmap shows the relative MIC values (calculated by dividing the MIC of the evolved line by that of the corresponding ancestor then performing $\log_{10}$-transformation) of 12 polymyxin B-resistant lines against ST and DT permeabilizer antibiotics used in this study. 3 independently evolved lines from each investigated species were studied, including *E. coli* (EC), *K. pneumoniae* (KP), *A. baumannii* (AB), and *P. aeruginosa* (PA). Gray color of the main heatmap indicates combinations not tested due to intrinsic resistance of ancestor strain. The additional heatmap to the right depicts genes mutated in each polymyxin B-adapted line and their involvement in specific cellular processes based on KEGG (Kyoto Encyclopedia of Genes and Genomes) and COG (Clusters of Orthologous Genes) databases. The distances between antibiotics were computed using the Euclidean distance method based on the log-transformed MIC values. The clustering was performed using the complete linkage method. Polymyxin B-resistant strains exhibited increased resistance to various ST permeabilizer antibiotics (*e.g.*, SPR206 (SPR) and additionally colistin (COL)), whereas DT permeabilizer antibiotics SCH79797 (SCH) and tridecaptin M152-P3 (TRD) maintained efficacy against polymyxin B-resistant strains. For detailed information on antibiotic and bacterial strain abbreviations, see Supplementary Data 1. Source data are provided as a Source Data file.

identified 14 non-redundant open reading frames (ORFs) in screens against DT permeabilizer antibiotics, with only three showing close sequence similarity to known resistance genes listed in the Comprehensive Antibiotic Resistance Database[54,55] (Supplementary Data 4).

These known resistance genes are associated primarily with antibiotic efflux mechanisms. The list includes *ramA* from the resistance-nodulation-cell division (RND) antibiotic efflux pump family (Supplementary Data 4), predominantly found in the clinical microbiome. The introduction of RamA into drug-susceptible *E. coli* strains induces a multi-drug resistant (MDR) phenotype, primarily by enhancing activity of the AcrAB-TolC efflux pump system and reducing porin production[56–58]. This aligns with the finding that overexpression of the AcrAB-TolC efflux system confers a mild, but significant reduction of susceptibility to SCH79797 in *E. coli* (Fig. 6). In the soil metagenomic library that was screened against POL7306, an additional ORF was identified that exhibited a 77% sequence similarity to CRP, a global regulatory gene that represses the *mdtEF* multidrug efflux genes[59].

We next examined the phylogenetic and ecological distribution of putative antibiotic resistance genes (ARGs) in natural isolates of *E. coli*, a species known for its wide ecological diversity and variable genomic content[60]. We analyzed genomic data from over 16,000 *E. coli* strains, categorized based on their habitats of isolation (agriculture, human, or wild animal hosts) and belonging to 11 different phylogenetic groups[61]. The analysis revealed that 48 of the putative ARGs identified in the metagenomic screens have close homologs in the *E. coli* genomes under study (Supplementary Data 4). These genes were generally present in a minority of the genomes. Intriguingly, however, we found that 14 ARGs (4 for SCH79797, 10 for POL7306) putatively conferring resistance to DT permeabilizers were conspicuously absent in all

habitats studied (Supplementary Fig. 8). Furthermore, upon analyzing these natural *E. coli* genomes, we found that none of them contained an ARG against any of DT permeabilizers (Fig. 8c). By sharp contrast, when the other three antibiotic groups were considered, up to 8.4% of the genomes carried at least one resistance gene.

### Killing kinetics of dual-target permeabilizers

We employed image analysis to quantify bacterial survival under fixed antibiotic concentrations (10xMIC for 4 hours) on *E. coli*, *A. baumannii*, *K. pneumoniae,* and *P. aeruginosa*. This analysis was conducted using a set of antibiotics to which these strains were sensitive (*e.g.*, MIC ≤ 4 μg/mL). Survival rate was quantified by calculating the ratio of the viable cells after antibiotic treatment and the total initial viable cell count. The analysis revealed that DT permeabilizers are among the antibiotics with the best killing kinetics. In particular, SCH79797 eliminated the entire bacterial population in all three bacterial species within 4 hours (Fig. 9a). Of note, colistin could eliminate the full bacterial population in *P. aeruginosa* only. We also calculated the resistance level (*i.e.*, median relative MIC) reached during laboratory evolution and survival rate under each antibiotic treatment and found a significant positive correlation between the two variables (Spearman's rank correlation coefficient $\rho = 0.37$, $p = 0.0019$, see "Methods").

To test the dose range under which resistance can evolve, we employed advanced microplating and automated image analysis. We quantified bacterial population survival at varying antibiotic concentrations[62] (see "Methods") in *E. coli*. Briefly, we generated dose-response curves and investigated how steeply the killing effect increases as the antibiotic concentration rises. In specific, the Hill coefficient representing each bacterial strain-antibiotic combination

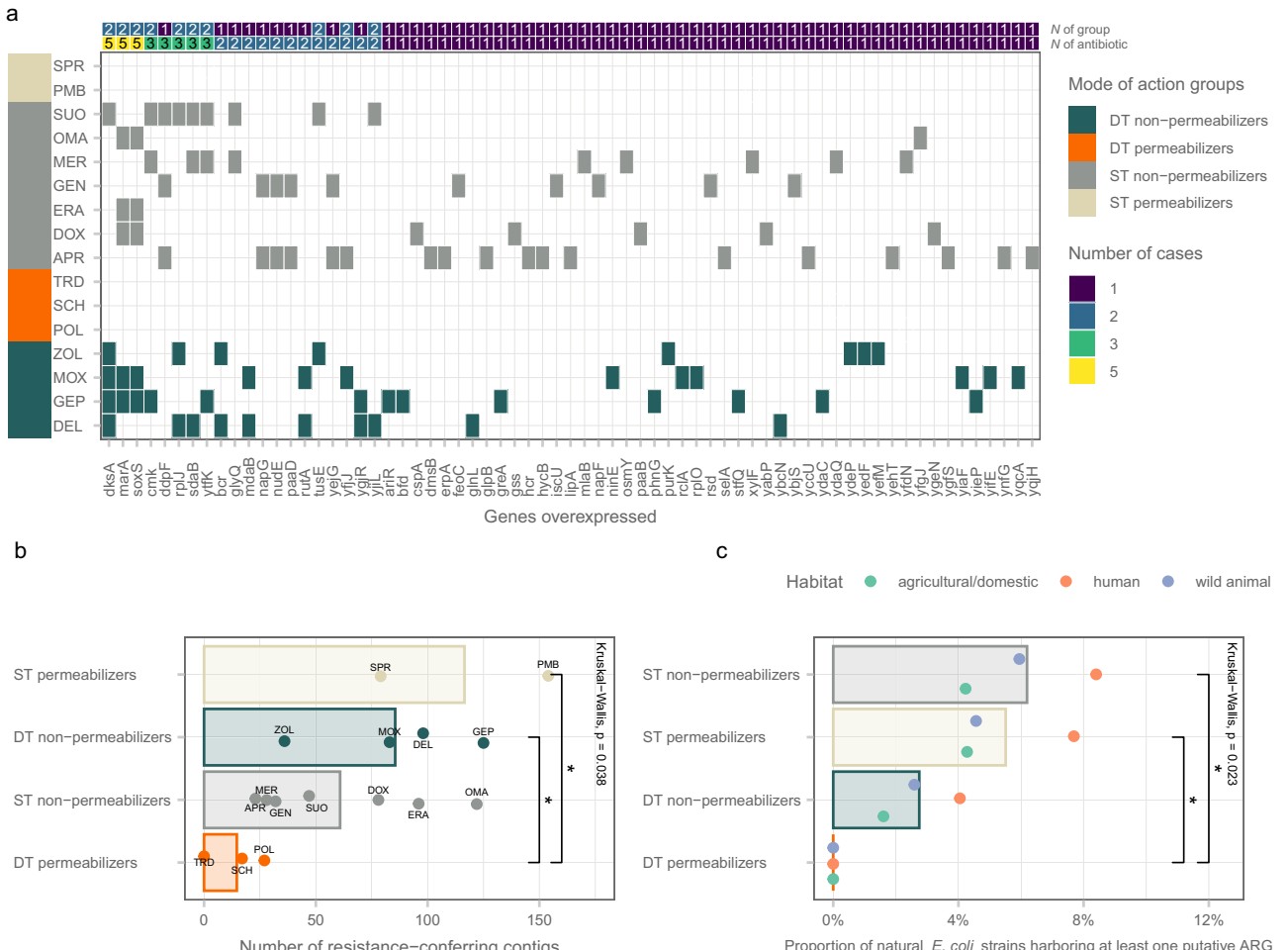

**Fig. 8 | Impact of gene overexpression and foreign DNA on antibiotic resistance. a** The heatmap depicts genes that reduce antibiotic susceptibility upon overexpression. The panel on the left denotes the mode of action group of the antibiotics. Panels from top to bottom correspond to the number of antibiotic groups and the number of antibiotics a given gene confers reduced susceptibility when overexpressed. Only genes with over 1% read coverage relative to the total coverage for each antibiotic were included (Supplementary Data 3). **b** Impact of foreign DNA on antibiotic resistance. The barplot shows the impact of foreign DNA segments, derived from functional metagenomics screens, on resistance to different antibiotics. Functional selection identified 1045 distinct antibiotic resistance-conferring DNA segments (contigs), while 4.2% of these were detected in screens against DT permeabilizers, including 17 for SCH79797 (SCH) and 27 for POL7306 (POL). The bars represent the mean of distinct contigs that provide resistance to each group of antibiotics. Individual data points reflect specific antibiotics within these groups. Statistical analysis was performed using two-sided Dunn's post-hoc test with Benjamini-Hochberg correction for multiple comparisons (* indicates $p = 0.0423$) following Kruskal-Wallis rank sum test ($n = 16$, chi-squared = 8.4475, df = 3, $p = 0.038$). **c** The prevalence of natural *E. coli* genomes containing putative antibiotic resistance genes (ARG). The barplot illustrates the proportion of natural *E. coli* genomes that contain at least one putative ARG across the four major groups of antibiotics. The bars represent the mean of the combined fraction of *E. coli* genomes with putative ARGs among habitats that provide resistance to each group of antibiotics. Individual data points reflect mean percentages of *E. coli* genomes with putative ARGs across antibiotics per habitat. Statistical analysis was performed using two-sided Dunn's post-hoc test with Benjamini-Hochberg correction for multiple comparisons (* indicates $p = 0.0364$) following Kruskal-Wallis rank sum test ($n = 12$, chi-squared = 9.4917, df = 3, $p = 0.023$). Notable differences were identified in the frequency of natural *E. coli* genomes that possess at least one putative ARG. For antibiotic abbreviations, see Table 1. Source data are provided as a Source Data file.

was calculated to quantify dose-kill kinetics. The analysis revealed that the dose-response curves for DT and ST permeabilizer antibiotics are remarkably steep, eradicating a substantial fraction of the bacterial population in a narrow dose range (Fig. 9b).

## Discussion

Given the urgent need for innovative solutions in antimicrobial therapy to combat the growing issue of resistance[63], our study evaluates the general principles that could guide the future development of antibiotics with limited resistance. In this study, we propose that to limit resistance development, new antibiotics should have dual modes of action and target membrane integrity. We show that only those antibiotics addressing both criteria exhibit limited resistance, while antibiotics with two intracellular targets remain susceptible to resistance development.

In particular, three preclinical antimicrobial agents, SCH79797, tridecaptin M152-P3, and POL7306, exhibited generally limited resistance development compared to all other antibiotics studied in this work. These antibiotic candidates induced membrane permeabilization and targeted a second cellular site, although their specific mechanisms of action varied (Table 1, Fig. 1, and Supplementary Fig. 1, 2). We found several factors that could hinder the evolution of resistance to these antibiotics. Unlike other antibiotics that are ineffective in one or more aspects and thus prone to resistance, dual-target permeabilizers simultaneously exhibit these features.

First, genomic mutations resulted in a generally limited elevation in resistance towards these types of antibiotics (Figs. 2a, b, and 3a). Cross-resistance between dual-target permeabilizers and other mode of action groups was rare (Fig. 5a, b, and Supplementary Fig. 5) due to the limited overlap in resistance mechanisms. Notably, polymyxin

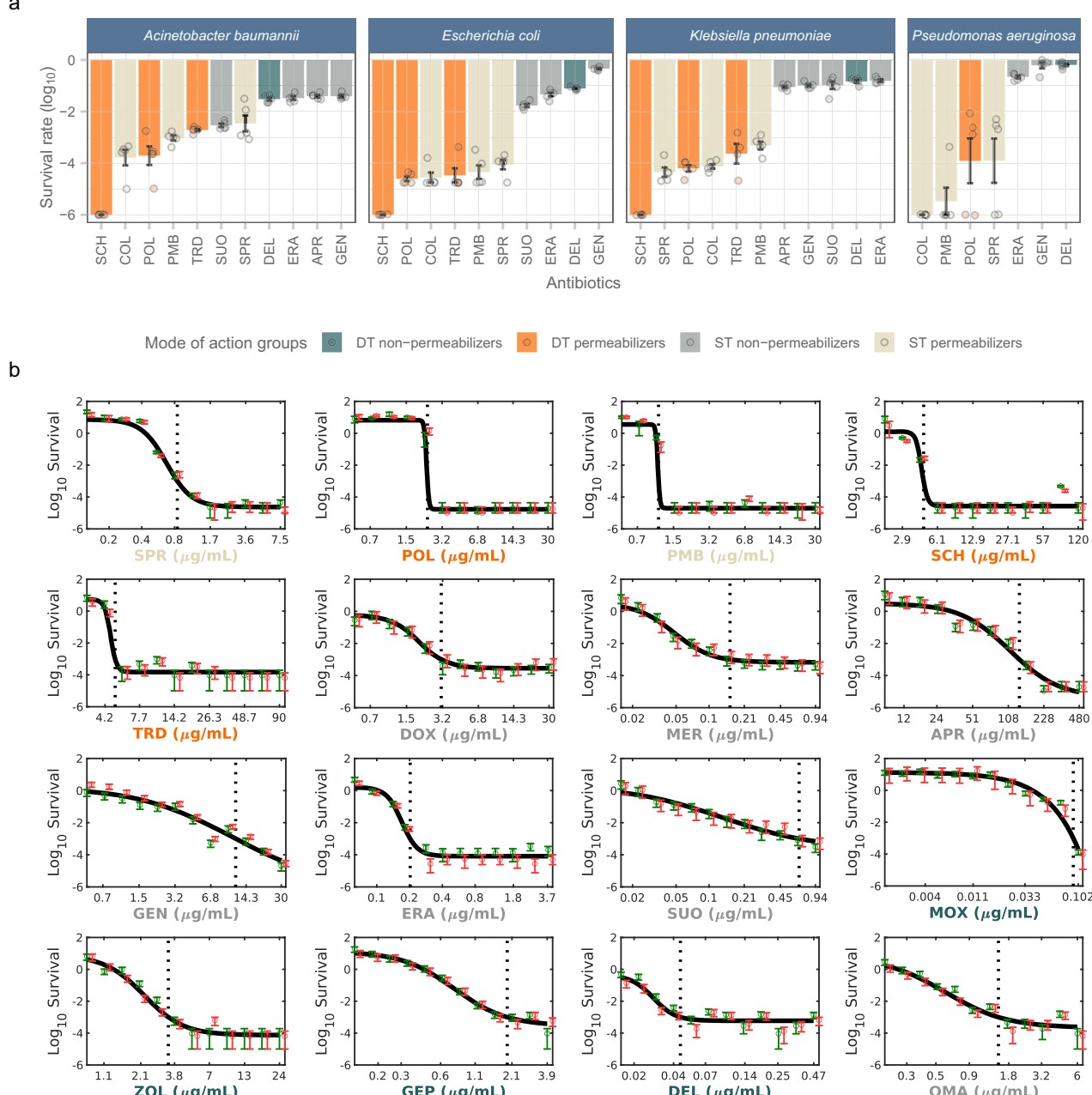

**Fig. 9 | Killing kinetics of the studied antibiotics. a** Bacterial survival rate in response to toxic antibiotic exposure. The survival rates of *E. coli, A. baumannii, K. pneumoniae,* and *P. aeruginosa* were assessed against antibiotics for which the strains demonstrated sensitivity (*e.g.*, MIC ≤ 4 μg/mL). This evaluation was conducted after a 4-hour treatment period using antibiotic concentrations equivalent to 10 times the MIC. The bars indicate the survival rate (log$_{10}$) compared to the untreated initial viable cell count (see "Methods"). Data are presented as mean values of 5 biological replicates ± SEM. The antibiotic group differentiation is denoted by color. DT permeabilizers, and especially SCH79797 (SCH) are among the antibiotics with the lowest bacterial survival rates. **b** Dose response curves of the studied antibiotics. The figure shows bacterial survival across different concentrations of antibiotics. Survival was estimated by measuring cell viability (colony-forming units/ml) of 2 biologically independent strains following a 4-hour exposure to different concentrations of 16 antibiotics on *E. coli* ATCC25922 (for

further details, see "Methods"). The antibiotic concentration that kills 99.9% of the *E. coli* population is indicated as a vertical dotted line (see Supplementary Data 5). Error bars represent standard error calculated from the colony counts by Poisson's model. A Hill function was fitted to the dose-response data from two biological replicates, producing sigmoidal curves (Supplementary Data 5). The Hill coefficient reflects how steeply the survival rate decreases in response to increasing antibiotic concentration (a lower value indicates lower survival). The log$_{10}$−transformed Hill coefficient for DT permeabilizers was significantly lower than for DT non-permeabilizers and ST non-permeabilizers (two-sided Student's *t*-test with Holm correction for multiple comparisons, *p* = 0.012 / 0.008, respectively), but statistically equivalent to that of ST permeabilizers (*p* = 0.218). *X*-axis label colors indicate mode of action groups. For detailed information on antibiotic and bacterial species, see Supplementary Data 1. Source data are provided as a Source Data file.

B-adapted lines exhibited no cross-resistance to DT permeabilizer antibiotics while being resistant to ST permeabilizers (Supplementary Fig. 6). This resistance pattern highlights the potential advantage of DT permeabilizers, as they appear less prone to promoting cross-

resistance through modification of the LPS of the bacterial membrane. Additionally, resistance could also be limited by the significant impact of resistance mutations on bacterial growth (Fig. 4), but further experiments are needed to resolve this issue.

Second, gene amplification is known to play a significant role in antibiotic resistance by increasing the expression of genes that confer resistance to antibiotics[44]. However, this strategy is not effective for antibiotics that permeabilize the bacterial cell membrane, probably because of the deleterious side effects of such alterations on membrane composition and structure (Fig. 8a).

Third, mobile resistance genes against such antibiotics are rare in human-associated microbiomes and clinically relevant pathogens. The absence of resistance-conferring DNA fragments against tridecaptin M152-P3 suggests a low risk of resistance. Additionally, the significantly lower diversity of SCH79797 and POL7306 resistance genes in both environmental and clinical samples indicate a reduced chance of resistance through horizontal gene transfer (Fig. 8b, c).

Fourth, dual-target permeabilizers exhibit strong bactericidal activity, characterized by a steep increase in efficacy within a narrow dose range compared to most other antibiotics (Fig. 9a, b). Other antibiotics, like polymyxin B and colistin, also effectively killed most bacterial populations, but simultaneously showed vulnerability to bacterial resistance. Therefore, future studies should investigate the relationship between antibiotic's killing efficacy and resistance evolution in more detail.

Our work indicates that typical resistance mechanisms, commonly found in bacterial populations, offer limited resistance to dual-target permeabilizers. We discuss each of them in turn.

Antibiotic resistance often arises through structural modifications of targeted proteins and cellular pathways. However, such mutations are notably absent in bacteria treated with dual-target permeabilizers. For instance, SCH79797, which binds to the bacterial channel MscL[19] and inhibits folate biosynthesis[17], showed no mutations in the corresponding genes in SCH79797-adapted bacteria. Similarly, no mutations emerged in the outer membrane protein BamA, the main target of POL7306[12]. Furthermore, although tridecaptin M152-P3 primarily binds to lipid II[13–15], no mutations were found in genes involved in lipid II biosynthesis or regulation[64]. Changes in the activity of major bacterial efflux pumps represent another common resistance mechanism. In the case of POL7306 and tridecaptin M152-P3, no mutations in efflux-related genes were detected in laboratory settings. However, SCH79797-adapted bacteria exhibited a small, but significant decline in susceptibility due to enhanced activity of the AcrAB-TolC efflux pump. Of note, no resistance via enzymatic inactivation was observed for dual-target permeabilizers in laboratory evolution (Supplementary Data 2) and functional metagenomic screens (Supplementary Data 4).

These findings contrast with the wide range of functionally distinct mutations and antibiotic resistance genes observed in response to treatments with dual-target topoisomerase antibiotics. These antibiotics target two homologous intracellular protein complexes, DNA gyrase and topoisomerase IV. The corresponding evolved lines exhibited a median antibiotic resistance level that is 128 times higher than the ancestor (Fig. 3a). Putative resistance mutations were observed regularly in targeted proteins and major efflux pumps[28]. In addition, functional metagenomic screens revealed that established genes involved in antibiotic target protection (e.g., QnR protein family)[65,66] provided resistance to several dual-target topoisomerase antibiotics (Supplementary Data 4). It is an open issue whether other non-permeabilizer dual-target antibiotics are also susceptible to resistance evolution.

Finally, it is important to contrast the resistance susceptibility of dual-target permeabilizers to polymyxin B and SPR206, which primarily bind to the lipid A portion of lipopolysaccharides (LPS) and thereby disrupt the bacterial membranes[26,27]. Resistance to these compounds readily occurs through LPS modifications and other cellular mechanisms that alter bacterial outer membrane composition. Dual-target permeabilizers POL7306 and tridecaptin M152-P3 have additional antibacterial activities unaffected by LPS-modifying mutations, resulting in limited resistance.

In sum, LPS is an antibiotic target prone to resistance formation. An alternative strategy to achieve membrane permeabilization is by binding to the mechanosensitive channel MscL, as exemplified by SCH79797. MscL is a promising new target for antibacterial development, as it is essential for bacterial survival, has a highly conserved protein sequence across bacterial species, including pathogens, and is absent in mammals. MscL channels are active throughout all stages of bacterial growth, including the stationary phase, and their function does not depend on cellular metabolism or energy[67]. Therefore, inhibitors targeting MscL channels should be effective against both stationary-phase cultures and dormant cells. Perhaps as a result, SCH79797 demonstrated exceptionally high antibacterial activity, successfully eliminating the entire bacterial populations within just four hours of treatment (Fig. 9a). Finally, MscL remains unmutated in response to SCH79797 stress. This pattern demands explanation as several other diverse MscL antagonists have been discovered but their susceptibility to bacterial resistance is unknown[18,19].

Our study investigated the effects of diverse antibiotics on bacterial resistance evolution to elucidate factors contributing to antibiotic efficacy. We identified key principles of antibiotic action that minimize resistance and proposed potential directions for future antibiotic development. However, further in vivo experiments are needed to enhance and validate the findings presented here. More generally, these promising results suggest a fruitful direction for future therapeutic applications, advocating for a focus on dual-target mechanisms that disrupt bacterial membranes while targeting additional cellular components. Our study focused on antibiotics effective against Gram-negative bacterial species with critical clinical importance. Therefore, it remains to be shown whether similar conclusions hold for antibiotics specific against infections caused by Gram-positive pathogens[10,68–71]. Our work also highlights the complexity of resistance mechanisms and the importance of a strategic, informed approach to antibiotic design and use.

## Methods
### Strains, antibiotics and media
This study focused on multiple bacterial strains, as follows. For the FoR and ALE experiments, 2 strains per species were chosen: *E. coli* ATCC 25922, *K. pneumoniae* ATCC 10031, *A. baumannii* ATCC 17978, and *P. aeruginosa* ATCC BAA-3107 as sensitive (SEN) strains, and *E. coli* NCTC13846, *K. pneumoniae* ATCC 700603, *A. baumannii* ATCC BAA-1605 and *P. aeruginosa* LESB58 as multi-drug resistant (MDR) strains. Functional metagenomic screens were performed with *E. coli* ATCC 25922 and *K. pneumoniae* ATCC 10031 strains. To determine the dose-kill properties of the different antibiotics, an *E. coli* strain (ATCC 25922) expressing a yellow fluorescent protein (pZS2R, YFP, CHL$^R$) and a red fluorescent protein (pZS2R, mCherry, CHL$^R$) was used.

Sixteen antibiotics were applied in this study from different antibiotic classes, including eleven newly developed antibiotics, which are in different phases of clinical trials and five conventional antibiotics with long clinical history from each antibiotic class, and categorized in four mode of action groups. For name, abbreviation and further details see Table 1. Antibiotics were custom synthesized or were purchased from several distributors (Supplementary Data 1). Upon preparation, each antibiotic stock solution was filter-sterilized and kept at −20 °C until usage.

Unless otherwise indicated, cation-adjusted Mueller-Hinton Broth 2 (MHB, Millipore) medium was used throughout the study, except for SCH79797 (SCH). To maximize antibacterial activity of SCH, based on prior experience with folate biosynthesis inhibitor antibiotics, Minimal Salt (MS) medium was used (1 g/L (NH$_4$)2SO$_4$, 3 g/L KH$_2$PO$_4$ and 7 g/L K$_2$HPO$_4$) supplemented with 1.2 mM Na$_3$C$_6$H$_5$O$_7$ × 2H$_2$O, 0.4 mM MgSO$_4$, 0.54 µg/mL FeCl$_3$, 1 µg/mL thiamine hydrochloride, 0.2% Casamino-acids and 0.2% glucose).

## Membrane permeabilization assay

The ability of antibiotics to permeabilize the outer membrane of *E. coli* ATCC 25922 was determined using the fluorescent probe, 1-N-phenylnaphthylamine (NPN, Merck)[72]. Cells were grown overnight until saturation in MHB then 200 μL of the overnight culture was used to inoculate 20 mL of MHB and the cells were grown to mid-logarithmic phase ($OD_{600}$ 0.4–0.6). Bacterial cultures were harvested by centrifugation at $1100 \times g$ for 10 min. Next, cells were washed and resuspended to an $OD_{600}$ of 0.4–0.6 in a buffer, containing 5 mM of HEPES (pH 7.0, Gibco™), 5 mM of glucose. Bacterial suspension (100 μL) with four times the MIC concentration of the tested antibiotics was added to the wells of a black 96-well plate (Revvity PhenoPlate™-96) and mixed with freshly prepared NPN solution (50 μL) and HEPES buffer (50 μL). Next, the mixture was left to equilibrate at room temperature for 15 min. Fluorescence was read immediately using a fluorescence plate reader (Biotek Synergy H1 microplate reader, $\lambda_{ex}$ = 350 nm, $\lambda_{em}$ = 410 nm). The wild-type *E. coli* treated with polymyxin B (final concentration 10 μg/mL) served as a positive control and fluorescence was calculated relative to the untreated control strain.

## Frequency-of-resistance assays

To estimate the frequency of spontaneous mutations that confer resistance in a microbial population, we conducted a frequency of resistance (FoR) assay following established protocols[10,31–33]. Approximately $10^{10}$ stationary-phase cells were plated onto MHB agar plates containing antibiotics at concentrations of 2×, 4×, 8×, and 20× the minimum inhibitory concentration (MIC). Prior to plating, bacteria were grown overnight in MHB medium at 37 °C and 250 RPM, then centrifuged and washed in MHB. All experiments were performed in triplicate, and plates were incubated at 37 °C for 48 hours. Total colony-forming units (CFUs) were determined by plating appropriate dilutions onto antibiotic-free MHB agar. Resistance frequencies were calculated by dividing the number of resistant colonies by the initial viable cell count. From the plates with the highest antibiotic concentration, ten colonies were selected for MIC determination and whole-genome sequencing.

## High-throughput laboratory evolution

An established protocol for adaptive laboratory evolution[46,73] was followed to propagate populations with the highest resistance levels. Starting with an antibiotic concentration that caused approximately 50% growth inhibition, ten parallel populations for each antibiotic-ancestor strain combination were grown for 72 hours at 37 °C with continuous shaking (300 RPM). MHB broth medium was used unless stated otherwise. After each incubation, 20 μl of each culture was transferred to four new wells containing fresh medium with increasing antibiotic concentrations (0.5×, 1×, 1.5×, and 2.5× the previous concentration) in a chessboard layout to monitor potential cross-contamination. Growth was monitored by measuring $OD_{600}$ using a Biotek Synergy 2 microplate reader, and only populations grown in the highest drug concentration with an $OD_{600} > 0.2$ were transferred further. The experiment continued for twenty transfers, resulting in 728 evolved lines.

## High-throughput MIC measurements

A standard serial broth microdilution technique was used to determine MICs, as suggested by the Clinical and Laboratory Standards Institute (CLSI) guidelines. A robotic liquid handling system was used to automatically prepare a 11 to 16-step serial dilution in 384-well microtiter plates (Greiner). A final concentration of $5 \times 10^5$ bacterial cells per mL were inoculated into each well containing 60 μL MHB medium. Bacterial cultures were incubated at 37 °C with continuous shaking (Infors shaker, 900 rpm with 3 mm throw) for 18 hours (2 replicates from each). Cell growth was monitored by measuring the optical density

($OD_{600}$ values, using Biotek Synergy microplate reader). MIC was defined as the antibiotic concentration of complete growth inhibition (*i.e.*, $OD_{600} < 0.05$). Relative MIC was calculated as follows:

$$\text{relative MIC} = \frac{\text{MIC}_{\text{evolved}}}{\text{MIC}_{\text{ancestor}}} \qquad (1)$$

## In vitro growth measurements

In this work we observed the growth phenotype of bacterial populations with available whole genome sequencing data from a previous study by assessing their growth at 37 °C in antibiotic-free MHB medium following established protocols[74]. We inoculated $5 \times 10^4$ cells from early stationary-phase cultures (prepared in MHB medium) into 60 μL of MHB medium in a 384-well microtiter plate (Greiner) and monitored bacterial growth for approximately 24 hours in a BioTek Synergy HL1 microplate reader. During the kinetic run, the optical density of each well was recorded at 600 nm ($OD_{600}$) every 5 minutes. Between the optical readings, the cultures were incubated at 37 °C under continuous shaking. Each bacterial variant and their corresponding wild types were measured in at least 9 replicates (3-3 biological in 3-3 technical). Finally, fitness was calculated from the obtained growth curves according to a previously described procedure[74]. Briefly, we calculated the area under the growth curve (AUC) from the beginning until 24 hours. AUC has been previously used as a proxy for fitness because it has the advantage of integrating multiple fitness parameters, such as the slope of the exponential phase (*i.e.*, growth rate) and the final biomass.

## Whole Genome Sequencing and Variant Analysis

To identify potential antibiotic resistance mutations, we selected 2 to 5 lines from the frequency-of-resistance (FoR) and adaptive laboratory evolution (ALE) experiments for whole genome sequencing. Resistant populations were grown overnight in antibiotic-free medium, and DNA was isolated using the GenElute Bacterial Genomic DNA Kit (Sigma), following the manufacturer's instructions. DNA was eluted in 120 μl of RNAse-free water over two elution steps, and 60 μl of the eluted DNA was further concentrated using the Zymo DNA Clean and Concentrator Kit. DNA concentration was measured with a Qubit Fluorometer and set to 1 ng/μl per sample. Sequencing libraries were prepared using the Nextera XT DNA library preparation kit (Illumina) and sequenced on an Illumina NextSeq 500 platform, generating $2 \times 150$ bp paired-end reads.

To identify and annotate variants, sequencing reads were mapped to their corresponding reference genomes using the Burrows-Wheeler Aligner (BWA)[75]. PCR duplicates were removed using Picard Mark Duplicates (see http://broadinstitute.github.io/picard) and reads with more than 6 mismatches were discarded. Variants (SNPs and INDELs) were called with Freebayes[76] and filtered using vcffilter[77], retaining only those with a quality score above 100. Rare variants were not excluded, but potential artefacts were manually inspected using IGV[78]. Variants were annotated with SnpEf, keeping only those absent in the ancestor. Mutations appearing in more than nine lines were excluded, as these were likely present in the ancestor. Those found in 6 to 9 lines were manually inspected, and mutations in repetitive regions (40 bp or longer) were removed. Lines with mutations in *mutL*, *mutS*, or *mutY* genes, or with more than nineteen mutations (identified as hypermutators), were also excluded.

For gene mutation analysis across strain backgrounds, we supplemented existing functional annotations for 8 bacterial strains. Nucleotide sequences and annotations for six strains (*E. coli* ATCC 25922, *K. pneumoniae* ATCC 10031, *A. baumannii* ATCC 17978, *P. aeruginosa* ATCC BAA-3107, *K. pneumoniae* ATCC 700603, and *A. baumannii* ATCC BAA-1605) were downloaded from the ATCC database, while genomic data for *P. aeruginosa* LESB 58 and *E. coli* NCTC13846

were obtained from NCBI (accession numbers FM209186.1 and NZ_UFZG00000000.1). All genes, including hypothetical ones, were functionally annotated using PANNZER2[79,80]. Gene sets were compared across strains by identifying orthologous groups using OrthoFinder (version 2.5.4)[81,82].

## Comparative Mutational Analysis

For this study, we performed a comparative mutational analysis to explore the genomic adaptations associated with different antibiotics. The analysis pipeline was implemented in R[83] (version 4.3.2), utilizing clusterProfiler package for pathway enrichment analyzes. Subsequent steps included an annotation process via the 'clusterProfiler' package[84], to establish KEGG pathway associations. The gene list was tailored for the organism *E. coli* K-12 MG1655. We also merged our dataset with additional functional annotations from COG database[85] to enhance our understanding of the potential resistance profiles.

## AcrAB efflux system on antibiotic resistance

For the efflux deficiency screen, the Δ*acrB* mutant of *E. coli* K12 BW25113[86] and its equivalent wild-type control strain with intact gene were utilized. In contrast, BW25113 carrying a multicopy acrAB plasmid (pUCacrAB)[87] and its control strain harboring an empty pUC118 plasmid were used to overexpress the acrAB efflux system. According to established protocols via P1 transduction, the suitable *acrB* deletion mutant (Δ*acrB* BW25113) was created[86], while the pUCacrAB plasmid was kindly provided by Kunihiko Nishino and Akihito Yamaguchi (Osaka University, Osaka, Japan). In a prior study, both strains were employed[88]. Since the copy number of the plasmid carrying the efflux pump is controlled by native promoters, external induction is unnecessary.

Antibiotic susceptibilities of these strains were evaluated using a modified version of the standard broth dilution method[89]. In 96-well microtiter plates containing fresh MHB medium, or supplemented MS medium in case of SCH79797, ten-step serial dilutions of an antibiotic from a stock solution were made (2 wells per antibiotic concentration per strain). The two wells contained only supplemented media without any antibiotics to suppress bacterial growth. Strains were incubated overnight at 37 °C with continuous agitation. Additionally, plasmid containing strains were grown in the presence of 100 μg/mL ampicillin. The culture was diluted to yield a cell suspension with $5 \times 10^5$ cells/mL and then inoculated into the wells of the microtiter plate. To determine the medium's background OD value, the first column of the 96-well plate contained solely MHB without inoculum. Prior to the measurement, the plates were incubated at 37 °C with continuous shaking at 900 rpm with 3 mm throw. $OD_{600}$ values were determined in a microplate reader after 18 hours of incubation (Biotek Synergy 2). The minimum inhibitory concentration (MIC) was determined as the lowest concentration of antibiotic at which the $OD_{600}$ values were less than 0.05 after subtracting the background value. The $\log_2$ of the relative MIC was computed using the following equation:

$$\log_2\left(\text{relative MIC}_{\text{pUCacrAB}}\right) = \frac{\text{MIC}_{\text{pUCacrAB}}}{\text{MIC}_{\text{pUC118}}} \qquad (2)$$

$$\log_2\left(\text{relative MIC}_{\Delta acrB}\right) = \frac{\text{MIC}_{\Delta acrB}}{\text{MIC}_{\text{wt}}} \qquad (3)$$

## Genome-wide overexpression library screening

Using MHB agar plates that included a concentration gradient of a given antibiotic, selections for resistance through overexpression were carried out[47]. To this end, we used the complete set of *E. coli* K-12 Open Reading Frame Archive (ASKA)[45] plasmid library (GFP minus) where each *E. coli* ORF is cloned into a high copy number expression plasmid (pCA24N-ORF_{GFP(-)}). Previously transformed *E. coli* K12 BW25113 strain bearing the plasmid pool ASKA was utilized in this work[46,90]. Frozen

aliquots of the pooled overexpression library were inoculated into 5 mL of fresh MHB containing chloramphenicol (20 μg/mL) and incubated at 37 °C until the optical density ($OD_{600}$) of suspension reached 0.8. Subsequently, as for preinduction, bacterial cells were exposed to 50 μM IPTG at 30 °C for 1 hour under constant agitation at 200 rpm. The suspension was then diluted to yield $5 \times 10^5$ cells in 50 μL and distributed using glass beads on gradient agar plates containing 100 μM IPTG, 10 μg/mL chloramphenicol, and the antibiotic concentration gradient. Inoculated plates were incubated for 24 hours at 37 °C in two replicates. For each antibiotic, a control plate containing the same number of cells harboring the empty plasmid (pCA24N-noORF, the plasmid without a cloned ORF) was prepared and used to determine the inhibitory zone of the antibiotic for cells lacking the ASKA plasmid. To isolate resistant clones from the libraries, sporadic colonies from the part of the plate that is distant from the inhibitory zone were washed together, as determined visually by comparing the inhibition zone on the control plate. Isolation of plasmid DNA was performed according to the manufacturer's instructions using the GeneJET Plasmid Miniprep Kit (Thermo Scientific™). To remove contamination caused by the genomic DNA, a mixture of two exonucleases was used. The isolated plasmid DNA samples were digested with 2.5 U of Exonuclease-I (20 U/μL) and Lambda-Exonuclease (10 U/μL) enzymes (Thermo Scientific™) for 1 μg plasmid DNA in Exonuclease I buffer. The reaction was carried out in ≤50 μL final volumes using a Thermal Cycler (Bio-Rad). The incubation period was 30 minutes at 37 °C followed by 15 minutes at 80 °C for inactivation. Purification of digested plasmid DNA samples was performed according to the manufacturer's instructions using the DNA Clean & Concentrator (Zymo Research) kit. Purified plasmids were subjected to sequencing using the Illumina platform.

## Overexpression library sequencing

Samples were processed with Nextera XT DNA Library Preparation Kit (Illumina, Cat. No. FC-131-1096). 1 ng DNA was used as the input volume recommended in the Nextera XT Sample Preparation Guide. Sequencing ready libraries were quality control checked by BioAnalyzer2100 instrument using High Sensitivity DNA Chip (Agilent Technologies USA, Cat. No. 5067-4626). NGS was carried out on the NextSeq 500 sequencing system with NextSeq 500/550 Mid Output Kit v2.5 (300 Cycles) chemistry (Illumina, Inc. USA, Cat. No. 20024905).

## ORF Identification from overexpression library

Two 150 bp sequences flanking the plasmid integration site ('E' denotes end, and 'S' the start) were identified from fastq files using the Smith-Waterman algorithm[91]. Reads containing sequences within 10 Hamming distance of the plasmid were filtered. Sequences E and S were searched in both R1 and R2 reads, including their reverse complements. At least 15 bp alignment to both the integration site and the ORF was required to retain a read. Identified reads were tagged with sequence information and orientation in the header, while excluded reads were replaced with "dummy" sequences to maintain file integrity. Filtered reads were mapped to the *E. coli* K-12 substr. MG1655 genome using BWA with a custom Burrows-Wheeler Transform (BWT) algorithm[75], as implemented by Delta Bio 2000 Ltd. (Szeged, Hungary). For open-source availability, this mapping step can be replicated using BWA. ORFs were identified between E and S sequences with continuous coverage and a minimum length of 15 bp. ORF annotation was performed based on the genome's GTF file, prioritizing protein-coding regions (CDS annotations). Reads containing E or S sequences were extracted from BAM files and validated based on the presence of stop codons and ORF length. ORFs were compared with CDS annotations, with identity scores reported as a percentage. Cases where ORFs exceeded the CDS length were also noted.

## Functional metagenomic screens

Resistance-conferring DNA fragments from the environment were identified through functional selection of metagenomic libraries. In previous work[30], metagenomic libraries were constructed to capture environmental and clinical resistomes, including: (i) river sediment and soil samples from seven antibiotic-polluted industrial sites near antibiotic production plants in India (anthropogenic soil microbiome), (ii) fecal samples from 10 European individuals (5 male, 5 female, age: 26-42) who had not taken antibiotics for at least one year (gut microbiome), and (iii) samples from a pool of 68 multidrug-resistant bacteria isolated from healthcare facilities or strain collections (see Supplementary Data 4). Detailed library construction methods are described by Apjok et al., 2023[30].

Permission for the fecal sample collection was obtained from the Human Investigation Review Board of Albert Szent-Györgyi Clinical Center, University of Szeged (registered under 72/2019-SZTE). Volunteer participants were selected on the basis of strict criteria that (1) they did not take any antibiotics for at least one yr before sample donation and (2) they are in a good health. These requirements are standard in the field and secure a bias-free comparison of the antibiotic resistomes in the healthy human gut microbiome. Informed consent was obtained from all participants.

Briefly, environmental DNA was isolated using the DNeasy PowerSoil Kit (Qiagen), and genomic DNA using the GenElute Bacterial Genomic DNA Kit (Sigma). The DNA was enzymatically fragmented, and fragments between 1.5–5 kb were selected. The length distributions of known antibiotic-resistance genes generally fall within this fragment size range, however some larger resistance elements may not be captured. Metagenomic inserts were cloned into a medium-copy-number plasmid with 10-nt barcodes (uptag and downtag) flanking each insert. Library sizes ranged from 2–6 million clones with an average insert size of 2 kb.

The libraries were introduced into *Klebsiella pneumoniae* ATCC 10031 via bacteriophage transduction (DEEPMINE)[30] and into *Escherichia coli* ATCC 25922 by electroporation. DEEPMINE uses hybrid T7 bacteriophage particles to optimize functional metagenomics in clinical bacterial strains. In a previous study[28], we made two modifications to the original protocol: (i) hybrid phages were generated using a T7 phage lacking the gp11-gp12-gp17 genes[92], and (ii) a new phage tail donor plasmid was used, which incorporated ΦSG-JL2 phage tail coding genes and the packaging signal region of the T7 phage[93].

Functional selections for antibiotic resistance were performed on MHB agar plates with a gradient of antimicrobial compounds[94,95]. Plates were incubated at 37 °C for 24 hours, and control plates with metagenomic plasmids lacking inserts were used to define the inhibitory zone. Resistant colonies, identified above the inhibitory zone, were collected for plasmid isolation. Metagenomic inserts in resistant clones were sequenced by two complementary methods: PCR amplification of the 10-nt barcodes flanking the metagenomic inserts, followed by Illumina sequencing, and full-length sequencing of inserts and barcodes via Nanopore.

## Annotation of antibiotic resistance genes (ARGs)

Consensus insert sequences from Nanopore sequencing were matched to their respective selection experiments using data from Illumina sequencing. First, Illumina sequencing reads were demultiplexed based on the 2×8 nt barcodes specific to each selection experiment, and then matched with the consensus insert sequences using the 10 nt barcodes unique to each metagenomic insert. To reduce redundancy and false matches, metagenomic contigs were filtered by: (i) retaining unique barcodes with the highest Nanopore read counts, and (ii) keeping contigs supported by at least 8 Nanopore and 5 Illumina reads.

Antibiotic resistance genes (ARGs) within these contigs were predicted using Prodigal v2.6.3[96] for open reading frame (ORF) prediction, followed by annotation of ORFs using the CARD[55] and

ResFinder[97] databases via blastx (NCBI BLAST v2.12.0[98]) with an E-value threshold of $10^{-5}$. ORFs were clustered at 95% identity and coverage using CD-HIT v4.8.1[99], retaining one representative ORF per cluster. Inserts were classified based on whether they contained ARGs, and if those ARGs were linked to the antibiotic used in the selection experiment.

Close orthologues of host-specific proteins were excluded by performing a blastp search against host proteomes (from UniProt) and removing ORFs with greater than 80% sequence similarity. The origin of the inserts was determined by searching Nanopore contigs within the NCBI Prokaryotic RefSeq Genomes database[100] using blastn (NCBI BLAST v2.12.0) and resolving taxonomic classifications using R[83] and the taxizedb package[101,102].

## Phylogenetic and geographic analysis of ARGs

Host type, geographic location, and phylogroup for a dataset of 16,272 *E. coli* genomes were determined based on previous work[61]. Initially, 26,881 *E. coli* genomes were retrieved from the NCBI RefSeq database in February 2022 and filtered to retain genomes with complete metadata. Clermont phylogrouping[103] was performed in silico using the EzClermont command-line tool[104], while host and location metadata were retrieved and categorized using Bio.Entrez utilities from Biopython v1.77. Genomes were grouped into three host categories: 'Human', 'Agricultural/Domestic animals', and 'Wild animals', by applying regular expressions to the 'host' field of biosample data. Geographic locations were divided into 20 subregions based on Natural Earth data[105].

A local blastp search was performed on this *E. coli* genome collection against a database of predicted antibiotic resistance gene (ARG) open reading frames (ORFs) identified in functional metagenomic screens. ARGs with at least 90% amino acid identity and 90% query coverage, and present in no more than 10% of the *E. coli* genomes, were analyzed further.

## Quantifying bacterial survival under antibiotic exposure

In this current study, the killing kinetics of each antibiotic was measured according to a previously established experimental protocol[62] with minor modifications. During this assay, the bacterial cultures were exposed to antibiotics in 96-well assay plates and their viability after 4 hours of exposure was quantified by two methods: i) microplating and colony counting or ii) fluorescent staining of cells and counting of the fluorescent units.

Specifically, these experiments were performed in the following steps: (1) Preparation of bacterial stocks. Single colonies from each strain were isolated and grown overnight in MHB medium at 37 °C, 160 rpm. In case of the fluorescently tagged *E. coli* strains, the medium was supplemented with 10 µg/mL chloramphenicol, to maintain the pZS2R plasmid. Overnight cultures were diluted 1:5000 and grown to an optical density at 600 nm ($OD_{600}$) of 0.25. These cultures were concentrated by centrifugation and small aliquots of 50 µL containing ~2.2 × 10^9 cells were frozen in glycerol (stored at −80 °C). (2) Preparing assay plates with antibiotics. 96-well plates were filled with 235 µL MHB and supplemented with antibiotics using an automated digital dispenser (D300e, Tecan). In the case of SCH79797, we used minimal media. For determining the dose-response curve for each antibiotic a 12-step dilution series (between 0.5 and 30 x MIC for each antibiotic) was prepared. For testing the clearance efficacy of each antibiotic at a single lethal dose, the 10x of the MIC for each antibiotic-strain pair were prepared in at least three replicates. (3) Antibiotic exposure. During the dose response assay, frozen cultures of the *E. coli* strains expressing YFP and mCherry were thawed and mixed at a 1:1 ratio, diluted 1:1000 in 30 mL MHB and incubated for 45 min at 37 °C and 160 rpm. During the single-dose assay, the frozen cultures of experimental strains (*E. coli*, *K. pneumoniae*, *A. baumannii*) were also thawed and diluted 1:1000 in 30 mL MHB and incubated at 37 °C to reach an

OD of 0.25 and 160 rpm. Following this incubation phase, these bacterial cultures were then aliquoted into the 96-well deep-well antibiotic plates (15 µL culture; total volume of 250 µL) at an inoculum density of about $10^7$ CFU. This inoculum density follows recommended clinical testing standards for killing assays[106]. No-cell and no-antibiotic wells were designated on each plate to control for contamination (one to two wells on each plate). (4) In the case of dose-response curves, we used microplating assay for quantification of bacterial viability by counting colony-forming units (CFU). Following a 4-hour antibiotic exposure step at 37 °C and 160 rpm, small aliquots (15 µL) were taken from each well, and serially diluted fivefold for plating. Based on previous observations[62], this 4-hour exposure step is sufficient for most antibiotics to achieve substantial population decline. We also plated the undiluted culture (dilutions of $5n$, where $n = 0$–$8$) and therefore added a wash step by centrifugation and resuspension to remove the drugs (three washes with 1200 µL PBS at 4,500 rpm for 15 min). Small microdrops (8 µL) from each dilution were then carefully microplated (Gilson 96-channel plate master) in one to four technical replicates onto omnitray single-well (Greiner) plates filled with 45 mL of agar with 70% MHB and incubated for 18 h at 37 °C. (5) Imaging. After incubation, the agar plates were imaged for YFP ($\lambda_{ex} = 500 \pm 10$; $\lambda_{em} = 606 \pm 94$) and mCherry ($\lambda_{ex} = 556 \pm 24$; $\lambda_{em} = 690 \pm 100$) with a custom-made automated macroscope device equipped with a Basler a2A840-45ucPRO camera. Photos were also taken without filters to image the nonfluorescent colonies. The number of bacterial colonies in each microplating spot was counted manually and the colony forming unit was calculated based on the dilution steps and the number of colonies present. To quantify viable cells while determining clearance efficacy of each antibiotic at a single lethal dose we used QUANTOM™ Viable Cell Staining Kit and QUANTOM Tx™ Microbial Cell Counter machine from Logos Biosystems following manufacturers recommendations.

### Statistics and Reproducibility

All in vitro measurements were performed with a minimum of two independent replicates to ensure the reliability of the results. Sample size was not predetermined using statistical methods, and no data points were excluded from the analysis. The experiments were not randomized, and investigators were not blinded during allocation or outcome assessment. Data processing and statistical analyzes were conducted using R programming language (version 4.3.2), with the tidyverse packages utilized for data analysis and visualization, and the stats and rstatix packages employed for statistical tests. KEGG pathway annotation of genes was performed using the clusterProfiler package. Matlab programming language was used to generate the dose response curves of the studied antibiotics. Unless specified otherwise, all statistical tests were two-tailed, and a $p$-value < 0.05 was considered indicative of statistical significance.

### Reporting summary

Further information on research design is available in the Nature Portfolio Reporting Summary linked to this article.

## Data availability

The outer membrane permeabilization, gene overexpression, cross resistance, AcrAB-TolC, killing kinetics data generated in this study and MIC fold change data after laboratory evolution used in this study are provided in the Supplementary Information/Source Data file. The Illumina reads for the gene overexpression experiments generated in this study have been deposited in the European Nucleotide Archive (ENA) database under accession code PRJEB80327. The Illumina reads and Nanopore contigs for whole genome sequencing and functional metagenomic screens used in this study are available in the European Nucleotide Archive (ENA) database under accession code PRJEB63210. Source data are provided with this paper.

## Code availability

The authors declare that all data cleaning and analysis associated with the current submission was performed using previously published methods, which applications were appropriately cited in the corresponding Methods sections. Mapping of ORFs from overexpression library was performed using BWA with a custom Burrows-Wheeler Transform (BWT) algorithm[75], as implemented by Delta Bio 2000 Ltd. (Szeged, Hungary). For open-source availability, this mapping step can be replicated using BWA. No further custom code was developed for the aforementioned purposes. Additional code underlying the figures featured are available from the corresponding authors upon request.

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

## Acknowledgements

Figure 1 was created in BioRender with a full licence for publication. Czikkely, M. (2025) https://BioRender.com/k95g583. This work was supported by:National Academy of Scientist Education Program of the National Biomedical Foundation under the sponsorship of the Hungarian Ministry of Culture and Innovation (CzM, LM). Cooperative Doctoral Program Scholarship of the Hungarian Ministry of Culture and Innovation (CzM, BB). The National Research, Development and Innovation Office, Hungary (NKFIH) grant FK-131961 (SJ). H2020-WIDESPREA-01-2016-2017-TeamingPhase2, GA:739593-HCEMM, EU's Horizon 2020 research and innovation program under grant agreement No. 739593 (SJ). Culture and Innovation of Hungary from the National Research, Development and Innovation Fund, financed under the TKP-2021-EGA-05 funding scheme (SJ). Lendulet "Momentum" program of the Hungarian Academy of Sciences (grant agreement LP2022-12/2022) (VL). EMBO Installation Grant (grant number 5709_2024) (VL). National Laboratory for Health Security Grant RRF-2.3.1-21-2022-00006 (BP). The European Union's Horizon 2020 Research and Innovation Program no. 739593 (BP). National Research Development and Innovation Office grants: 'Élvonal' Program KKP 129814 (BP). ERA-NET JPIAMR-ACTION (BP). National Laboratory of Biotechnology Grant 2022-2.1.1-NL-2022-00008 (CP, BP). National Research, Development and Innovation Office K146323 (CP). The European Research Council ERC-2023-ADG 101142626 FutureAntibiotics (CP).

## Author contributions

E.M., M.S.Cz., and C.P. conceptualized the project. E.M., M.S.Cz., P.Sz., V.L., and C.P. developed the methodology. E.M., M.S.Cz., P.Sz., Z.F., and V.L. conducted the formal analysis. E.M., M.S.Cz., P.Sz., L.D., E.K., L.M., T.K., B.B., and V.L. carried out the investigation. L.D., A.D., Sz.J., B.P., and V.L. provided resources. E.M., M.S.Cz., P.Sz., Z.F., G.G., and V.L. curated the data. E.M., M.S.Cz., and C.P. wrote the original draft, and all authors reviewed and provided feedback on the manuscript. E.M., M.S.Cz., and Z.F. handled visualization. A.D. and C.P. managed project administration. C.P. supervised the research and acquired funding.

## Funding

## Competing interests

Authors declare no competing interests.
