## [Peer Review File · Nature Communications]

Exploring the principles behind antibiotics with limited resistance

Corresponding Author: Dr Csaba Pál

Version 0:

Reviewer comments:

Reviewer #1

(Remarks to the Author)

It is extremely needed the development of new antibiotics with limited resistance. That is what Maharramov et al analyze in this beautiful piece of work: "Exploring the principles behind antibiotics with limited resistance". They propose looking for antibiotics with dual modes of action, one of the objectives being a non-mutable target. The hypothesis proposed by Csaba Pál Lab: "to limit resistance development, new antibiotics should have dual-modes of action and specifically target membrane integrity" sounds very reasonable. They state -and data support that statement- that "antibiotics addressing both criteria exhibit limited resistance" and that "antibiotics with two protein-encoded intracellular targets remain susceptible to resistance development". What I however think that data do not -completely- support is that "ONLY THOSE antibiotics addressing both criteria exhibit limited resistance". I think that Maharramov et al may be overestimating the effectiveness of DT- compared to ST-permeabilizers (at least for some particular bacterial species), something I mention below, as one of the most important points to address. Overall, the study raises very important new aspects useful for the development of antibiotics with limited drug resistance, something very necessary. I therefore recommend the publication of this article once some aspects have been addressed.

Major points:

- Supplementary Table 2, FoR data: For some bacterial strains, there is no advantage in using DT- over ST-permeabilizers regarding the possible selection of resistant mutants (*K. pneumoniae* ATCC 10031 SCH vs PMB; *E. coli* NCTC 13846 SPR vs PMB; *P. aeruginosa* ATCCBAA 3107 POL vs PMB or SPR; ALE data, *E. coli* NCTC 13846 SPR vs PMB). In fact, the use of the first may be riskier in occasions (i.e. possible selection of mutations in genes encoding the AcrAB-TolC efflux pump by SCH79797 and cross-resistance mechanisms to antibiotics from other families). These particularities at the species or strain level should be mentioned, since they could be useful by expanding the range of possible therapeutic options in these pathogens. In this respect, MICs for some antibiotics is missing in some species (f.e. DT-permeabilizers in *P. aeruginosa* and *k. pneumoniae*) and it makes difficult the interpretation and generalization of their results. If the authors have a reasonable explanation for having ruled out these antibiotics in these species, such as that they are intrinsically resistant, they should comment on it at the beginning of the results section, to facilitate understanding. I suggest highlighting the species where this conclusion ("only those...") is clearly supported and those for which the use of ST-permeabilizers seems a reasonable option.

- Having in mind the heterogeneity observed across bacterial species in ALE assays and that only a DT-permeabilizer (the most similar to polymyxin B) was analyzed in *P. aeruginosa*, I am not sure that it is correct to draw general conclusions including the data on this pathogen. It would be necessary, at least, to state this limitation and avoid making such a strong statement.

- Regarding the doubts that I expressed about the existence of an indubitable advantage in using DT- over ST-permeabilizers, the results of Maharramov et al indicate that both, DT- and ST-permeabilizers are the antibiotics with the lowest bacterial survival rates (Fig 7). While it is true that SPR (DT-permeabilizer) shows the lowest bacterial rate in the three species analysed, the other ST- and DT-permeabilizers tested show similar effects on survival rates. In fact, "dose-response curves for DT- and ST-permeabilizer antibiotics are remarkably steep". These results support that the statement "only those antibiotics addressing both criteria (DT-permeabilizers) exhibit limited resistance" is not entirely faithful to reality. That is why I suggest to the authors lower that statement and give more importance to the results obtained in permeabilizing

antibiotics, whether they have one or two targets. Finally, I missed the assay of survival rates in *P. aeruginosa* using the permeabilizers drugs included in FoR and ALE assays (PMB, SPR, POL), in addition to the classical antimicrobial peptide, COL. If there is an advantage of ST- over DT-permeabilizers (or if they are equally effective) in this bacterial species, it would make sense to state that the use of ST-permeabilizers seems a reasonable option against this pathogen.

Minor points:

- Maharramov et al analyze cross-resistance of polymyxin (ST-permeabilizer) and DT-permeabilizer resistant lines and find that they exhibit cross-resistance to ST-permeabilizers and antibiotics for other families, respectively. In my view, the fact that ST-permeabilizers select for resistance to other ST-type antimicrobial peptides but do not do so against the other three types of antibiotics is an advantage over the type of cross-resistance selection exerted by DT-permeabilizers. Perhaps this could also be discussed throughout the manuscript.

- Since the over-production of AcrAB-TolC efflux pump (able to extrude the DT-permeabilizer SCH79797) is frequently selected into the clinics, I think that SCH79797 would not be the best antibiotic to use for *E. coli* infections. However, it should be recommended in *Klebsiella* and *Acinetobacter*. This clarification at the species level would seem appropriate to me.

- You indicate in the discussion section that "MscL remains unmutated in response to SCH79797 stress" and that "this pattern demands explanation". I think the most reasonable explanation is given a few sentences above: "it is essential for bacterial survival". So, I agree that it would be very interesting to look for new inhibitors of this channel, especially those that cannot be expelled by efflux pumps.

Reviewer #2

(Remarks to the Author)

Comments to Maharramov et al. Nature Communications

This manuscript by Maharramov et al. is a very nice, interesting, important, useful and complete study of how to design antibiotics. The authors concluded that dual-target antibiotics, where one of the targets is membrane integrity, are much less prone to resistance than single-target antibiotics or even dual-target antibiotics where none of the targets is the membrane. I raised a few points I hope the authors can answer and correct.

Major points:

1) In a few places, sometimes the text gives us the (wrong) idea that there is some kind of Lamarckism (or mutation induction) in the process of antibiotic resistance. Since Luria and Delbrück (1943), we know that this is a typical process of (Darwinian) Natural Selection. Therefore, I think the authors should ameliorate a few sentences. For example, in the Abstract, line 32, the authors wrote "Third, the timeframe for resistance evolution was brief, owing to the rapid eradication of bacterial populations upon toxic antibiotic exposure.". As Luria and Delbrück and many others showed, resistant mutants were already resistant "before" they were in contact with the stress factor. Bacteria did not become resistant "because" they are in contact with the stress factor. That sentence refers to their beautiful experiments 7a and 7b. Yes, dual-target permeabilizers kill faster than the other antibiotic types. Still, this should be viewed as a lucky coincidence (faster killers are also the ones where resistance opportunities are rarer). It should not be considered an explanation for the inability of bacteria to "find" resistance. The paragraph in lines 342-345 (Discussion section) should also be clearer.

2) According to my previous point, I suggest that the authors comment on this coincidence.

3) Figures 2c, 2d, 6b and 6c are essential in this manuscript. Kruskal-Wallis analyses are insufficient because this test tells us that the groups are different, not necessarily the DT-perm antibiotics. For example a post-hoc test would be important.

4) I am also worried about a point in Methods (Lines 432-432) and then the interpretation of results. The authors write, "Unless otherwise indicated, cation-adjusted Mueller-Hinton Broth 2 (MHB, Millipore) medium was used throughout the study, except for SCH79797 (SCH). To maximize antibacterial activity of SCH, based on prior experience with folate biosynthesis inhibitor antibiotics, Minimal Salt...". Could that be the reason why this DT-perm was so efficient, fast-killer, etc? Could the results of this manuscript be an artifact of the medium? This should be explained.

Minor Points:

5) Line 506: in the 96-well plates, did the authors used MHB for the DT-perm SCH?

6) I also ask the authors to elaborate more on their discussion. Why is permeabilization so important? Why not just DT-non-permeabilizers?

7) In addition to the previous point, I would like to ask for a small discussion about what the authors would expect with a DT-permeabilizer where both targets are the membrane.

8) Lines 181-185 and Figure 4a. I made a (too) strong effort to understand these lines and figure 4a. Then I understood it, but I suggest the authors to explain better (set size, etc, etc).

9) Lines 211-212: in 12 strains?

10) Lines 275-278. The Kruskal-Wallis statistics is not testing what the sentence says (unless the authors do a post-hoc test).

11) Lines 275-280. This is a nice point, but could it be because DT-perm. are new antibiotics? Could you comment on this in the Discussion section?

12) Lines 283-286: again the "Lamarckism" or mutation induction issue I raised in my first point. Please note that Windels et al. (ref 62) refers to "Bacterial Persistence," as giving more opportunities to find resistance because, in the absence of antibiotics, they grow again, etc., and bacteria can mutate and become ready the next time the drug appears. It is NOT because bacteria have a wider window to "find" resistance.

13) Lines 342-349: again as in point 1 and 12

14) Figure 7b: I ask the authors to mark the DT-perm drugs to make it clearer to the reader.

Reviewer #3

(Remarks to the Author)

Review Maharramov et al, submitted to Nature Communications.

This is a comprehensive work on the interface between microbial evolution and antibiotic development. The manuscript is generally well written, and the vast amounts of data are nicely presented in a clear way. The data presented suggest that dual targeting can limit in vitro resistance evolution when membrane integrity is included as a target.

Points for discussion:

I have no major scientific concerns about the presented work- it is comprehensive and very impressive. As a general structural comment I believe the manuscript would benefit from being even more clear with respect to whether data were obtained from the mentioned work soon to be published in Nature Microbiology- and reanalyzed here. With the current version of ref 28 only present at bioRxiv I really would not know (in full awareness that the final published version could differ). To that end, another suggestion would be to bring the other paper up in the discussion to highlight contrasting points..

How certain can the authors be about the distinct classes of non-permeabilizers? Recent data suggest that classical non-permeabilizing drugs may interact with the membrane (<https://doi.org/10.1016/j.bbamem.2020.183448>).

FoR analyses: I could not find the actual frequencies anywhere- I also checked in the bioRxiv manuscript (Daruka et al)- presenting the frequency data of mutations at different concentrations would be interesting to see

Line 188: “..to Dt permeabilizers..” right?

Mobile resistance (from line 242)- these experiments were limited by 5 kb inserts- would it not be worth including this limitation explicitly? Vancomycin resistance for example is encoded by an 10,8 kb transposon.

Include materials and methods in main text and not separate Supplementary Information

Fig. 2d. boxes missing for Pa and Ab

Line 349: include PK/PD considerations as well as more in vivo like conditions to fully evaluate resistance potential?

Reviewer #4

(Remarks to the Author)

Version 1:

Reviewer comments:

Reviewer #1

(Remarks to the Author)

In this revised version, the authors have substantially reinforced the conclusions of the article in line with my concerns and they have carried out the experiment I suggested. I have nothing new to add and I would like to congratulate the authors for this nice piece of work, which is really necessary to guide the design of antibiotics less prone to resistance.

Reviewer #2

(Remarks to the Author)

The authors have carefully addressed all concerns.
Thank you

Reviewer #4

(Remarks to the Author)

Response to the reviewers

REVIEWER COMMENTS

Reviewer #1 (Remarks to the Author):

It is extremely needed the development of new antibiotics with limited resistance. That is what Maharramov et al analyze in this beautiful piece of work: “Exploring the principles behind antibiotics with limited resistance”. They propose looking for antibiotics with dual modes of action, one of the objectives being a non-mutable target. The hypothesis proposed by Csaba Pál Lab: “to limit resistance development, new antibiotics should have dual-modes of action and specifically target membrane integrity” sounds very reasonable. They state -and data support that statement- that “antibiotics addressing both criteria exhibit limited resistance” and that “antibiotics with two protein-encoded intracellular targets remain susceptible to resistance development”. What I however think that data do not -completely-support is that “**ONLY THOSE** antibiotics addressing both criteria exhibit limited resistance”. I think that Maharramov et al may be overestimating the effectiveness of DT- compared to ST-permeabilizers (at least for some particular bacterial species), something I mention below, as one of the most important points to address. Overall, the study raises very important new aspects useful for the development of antibiotics with limited drug resistance, something very necessary. I therefore recommend the publication of this article once some aspects have been addressed.

Major points:

- Supplementary Table 2,

FoR data: For some bacterial strains, there is no advantage in using DT- over ST-permeabilizers regarding the possible selection of resistant mutants (*K. pneumoniae* ATCC 10031 SCH vs PMB; *E. coli* NCTC 13846 SPR vs PMB; *P. aeruginosa* ATCC BAA 3107 POL vs PMB or SPR; ALE data, *E. coli* NCTC 13846 SPR vs PMB).

In fact, the use of the first may be riskier in occasions (i.e. possible selection of mutations in genes encoding the AcrAB-TolC efflux pump by SCH79797 and cross-resistance mechanisms to antibiotics from other families). These particularities at the species or strain level should be mentioned, since they could be useful by expanding the range of possible therapeutic options in these pathogens.

Thank you for raising this issue. To address the concern, we have generated two additional figures (Extended Data Fig. 3-4) and conducted corresponding statistical analyses. These new data clarify the strain-specific cases in the frequency of resistance (FoR) and the adaptive laboratory evolution (ALE) experiments, where the differences in the level of resistance (relative MIC) between the DT and ST permeabilizer antibiotics are thoroughly noted.

We modified the main text as follows:

Lines 147-177:

“In all bacterial species, resistance typically developed quickly in the three control groups, including ST permeabilizer, DT non-permeabilizer and ST non-permeabilizer antibiotics. By contrast, there was a low probability of resistance development against DT permeabilizers. In particular, the average increment in resistance level was less than fourfold in populations exposed to POL7306, tridecaptin M152-P3, and SCH79797 stresses (Fig. 2a and Supplementary Table 2).

In stark contrast, resistance levels increased by more than 128-fold for polymyxin B in A. baumannii multidrug-resistant (MDR) strain and SPR206 in E. coli and K. pneumoniae sensitive strains (Extended Data Fig. 3). Notably, no resistant mutants emerged during this assay for polymyxin B in A. baumannii and K. pneumoniae sensitive strains, nor in the multidrug-resistant E. coli strain. However, the underlying FoR assay cannot detect very rare mutations and combinations thereof, and hence they may underestimate bacterial potential for resistance³². Therefore, it is essential to demonstrate how long-term exposure to these antibiotics affects resistance evolution.

Using the same set of bacterial strains, we initiated adaptive laboratory evolution with the aim to maximize the level of antibiotic resistance in the populations achieved during a longer, but fixed time period²⁸. As expected, resistance levels were generally much higher than those observed in FoR assay.

Reassuringly, the level of resistance against DT permeabilizers was significantly lower in comparison to all other antibiotic groups (Fig. 2b, and Supplementary Table 2). For instance, after 60 days of evolution, resistance levels increased by a maximum of fourfold in A. baumannii and in the E. coli sensitive strain exposed to SCH79797. By contrast, polymyxin B-resistant lineages displayed over 1024-fold increments in resistance levels.

Of note, when analysing all antibiotic pairs on a strain-specific basis, DT permeabilizers consistently exhibited significantly lower median resistance levels than ST permeabilizers in 69% of the cases (Extended Data Fig. 4). Importantly, no instances were detected where an ST permeabilizer resulted in a lower relative MIC compared to any DT permeabilizer. Furthermore, the capacity to develop resistance against POL7306 and SCH79797 displayed significant heterogeneity across bacterial species (Fig. 2d; Kruskal-Wallis rank-sum test $p = 0.000011$ and $p = 0.0027$, respectively). For example, A. baumannii showed elevated resistance to POL7306 only, whereas K. pneumoniae exhibited increased resistance to both POL7306 and SCH79797 (Fig. 2d).”

Strain	Strain abbreviation
E. coli ATCC 25922	EC-SEN
E. coli NCTC 13846	EC-MDR
K. pneumoniae ATCC 10031	KP-SEN
K. pneumoniae ATCC 700603	KP-MDR
A. baumannii ATCC 17978	AB-SEN
A. baumannii ATCC BAA-1605	AB-MDR
P. aeruginosa ATCC BAA-3107	PA-SEN
P. aeruginosa LESB 58	PA-MDR

Extended Data Fig. 3. Resistance levels following the frequency of resistance assay across bacterial strains. The level of resistance was measured as the fold change in relative MIC between evolved and ancestral strains. Data are grouped by bacterial strain (*A. baumannii* (AB), *E. coli* (EC), *K. pneumoniae* (KP), and *P. aeruginosa* (PA)) and categorized as either multidrug-resistant (MDR) or sensitive (SEN). Each data point represents a laboratory evolved line, and boxplots show the median, first and third quartiles, with whiskers indicating the 5th and 95th percentiles. DT permeabilizers generally resulted in lower or equal resistance levels compared to ST permeabilizers. Statistical analysis was performed using Dunn's post-hoc test with Benjamini-Hochberg correction for multiple comparisons (**/ * indicates $p < 0.01/ 0.05$). For detailed information on antibiotic and bacterial species, see Supplementary Table 1. For mode of action groups, refer to Table 1.

Extended Data Fig. 4. Resistance levels following adaptive laboratory evolution across bacterial strains. The level of resistance was measured as the fold change in relative MIC between evolved and ancestral strains after ~60 days of adaptive laboratory evolution. Data are grouped by bacterial strain (*A. baumannii* (AB), *E. coli* (EC), *K. pneumoniae* (KP), and *P. aeruginosa* (PA)) and categorized as either multidrug-resistant (MDR) or sensitive (SEN). Each data point represents a laboratory evolved line, and boxplots show the median, first and third quartiles, with whiskers indicating the 5th and 95th percentiles. DT permeabilizers resulted in lower resistance levels compared to ST permeabilizers, though significant strain-specific variations were observed. Statistical analysis was performed using Dunn’s post-hoc test with Benjamini-Hochberg correction for multiple comparisons (****/ ***/ **/ * indicates $p < 0.0001/ 0.001/ 0.01/ 0.05$). For detailed information on antibiotic and bacterial species, see Supplementary Table 1. For mode of action groups, refer to Table 1.

In this respect, MICs for some antibiotics is missing in some species (f.e. DT-permeabilizers in *P. aeruginosa* and *k. pneumoniae*) and it makes difficult the interpretation and generalization of their results. If the authors have a reasonable explanation for having ruled out these antibiotics in these species, such as that they are intrinsically resistant, they should comment on it at the beginning of the results section, to facilitate understanding. I suggest highlighting the species where this conclusion (“only those...”) is clearly supported and those for which the use of ST-permeabilizers seems a reasonable option.

Thank you for your comment. We have now clarified the exclusion of certain antibiotics from the analysis, particularly where decreased susceptibility (e.g., $> 4 \mu\text{g/mL}$) was observed prior to laboratory evolution. Additionally, in response to your earlier comments, we have expanded on the strain-specific evolution of DT and ST permeabilizers to provide more context (see above).

Now in lines 140-144, we state:

“Antibiotic-strain combinations exhibiting decreased initial susceptibility (e.g., $> 4 \mu\text{g/mL}$) were excluded from the experiments (see Supplementary Table 1). For all other combinations, approximately 10^{10} bacterial cells were exposed to one of each antibiotic on agar plates for two days at concentrations where the given strain is susceptible.”

- Having in mind the heterogeneity observed across bacterial species in ALE assays and that only a DT-permeabilizer (the most similar to polymyxin B) was analyzed in *P. aeruginosa*, I am not sure that it is correct to draw general conclusions including the data on this pathogen. It would be necessary, at least, to state this limitation and avoid making such a strong statement.

We fully agree with your observation. Due to *P. aeruginosa*'s decreased susceptibility (e.g., > 4 µg/mL) to other DT permeabilizers tested, we were limited to using POL7306 as the only viable DT permeabilizer for this species. As a result, we understand that drawing broad conclusions from data in this pathogen may not be appropriate, and we have now added a statement to the manuscript to acknowledge this limitation.

We have included the following limitation in the Results section:

Lines 177-182:

*"It is important to note that due to *P. aeruginosa*'s decreased initial susceptibility (e.g., > 4 µg/mL) to other DT permeabilizers, only POL7306 was analyzed in this species. Notably, resistance level reached after adaptive laboratory evolution was significantly lower compared to ST permeabilizers (Extended Data Fig. 4, Dunn's post-hoc test with Benjamini-Hochberg correction, $p < 0.01$). However, further studies, including a broader range of DT permeabilizers, are needed to validate this result."*

- Regarding the doubts that I expressed about the existence of an indubitable advantage in using DT- over ST-permeabilizers, the results of Maharramov et al indicate that both, DT- and ST-permeabilizers are the antibiotics with the lowest bacterial survival rates (Fig 7). While it is true that SPR (DT-permeabilizer) shows the lowest bacterial rate in the three species analysed, the other ST- and DT-permeabilizers tested show similar effects on survival rates. In fact, "dose-response curves for DT- and ST-permeabilizer antibiotics are remarkably steep".

These results support that the statement "only those antibiotics addressing both criteria (DT-permeabilizers) exhibit limited resistance" is not entirely faithful to reality. That is why I suggest to the authors lower that statement and give more importance to the results obtained in permeabilizing antibiotics, whether they have one or two targets.

Several lines of evidence indicate that DT permeabilizers are less prone to resistance compared to ST permeabilizers.

During laboratory evolution (ALE), resistance levels were lower in 69% of the tested antibiotic-strain combinations. Importantly, there were no cases where DT permeabilizers showed higher resistance than ST permeabilizers. Notably, up to 1024-fold increments in resistance levels were observed for a ST-permeabilizer, polymyxin B.

Mobile resistance genes against DT permeabilizer were rare in human-associated microbiomes and clinically relevant pathogens, while they were abundant against ST permeabilizers.

Furthermore, polymyxin B-adapted lines exhibited no cross-resistance to DT permeabilizer antibiotics while being resistant to ST permeabilizers.

As regards the link between survival rates the dose-response curves, we write in the discussion:

Lines 372-376:

“Other antibiotics, like polymyxin B and colistin, also effectively killed most bacterial populations, but simultaneously showed vulnerability to bacterial resistance. Therefore, future studies should investigate the relationship between antibiotic’s killing efficacy and resistance evolution in more detail.”

Finally, I missed the assay of survival rates in *P. aeruginosa* using the permeabilizers drugs included in FoR and ALE assays (PMB, SPR, POL), in addition to the classical antimicrobial peptide, COL. If there is an advantage of ST- over DT-permeabilizers (or if they are equally effective) in this bacterial species, it would make sense to state that the use of ST-permeabilizers seems a reasonable option against this pathogen

We conducted the requested experiments and observed that POL7306 demonstrated killing kinetics comparable to those of other ST permeabilizers, such as polymyxin B and SPR206, consistent with findings in other species. Notably, in this strain colistin completely eradicated the entire population after 4 hours of treatment. However, we cannot make the suggested recommendations as in our adaptive laboratory evolution (ALE) experiments showed high levels of ST permeabilizer resistance in *P. aeruginosa* (e.g., 32-fold increase in polymyxin B and SPR206 resistance level).

Fig. 7a. Bacterial survival rate in response to toxic antibiotic exposure. The survival rates of *E. coli*, *A. baumannii*, *K. pneumoniae*, and *P. aeruginosa* were assessed against antibiotics for which the strains demonstrated sensitivity (e.g., MIC ≤ 4 µg/mL). This evaluation was conducted after a 4-hour treatment period using antibiotic concentrations equivalent to 10 times the MIC. The bars indicate the survival rate (log₁₀) compared to the untreated initial viable cell count (see Methods), with whiskers indicating the standard error based on 5 biological replicates. The antibiotic group differentiation is denoted by colour. DT permeabilizers, and especially SCH79797 (SCH) are among the antibiotics with the lowest bacterial survival rates. For detailed information on antibiotic and bacterial species, see Supplementary Table 1. For antibiotic groups, refer to Table 1.

We modified the corresponding section as follows:

Lines 313-323:

“We employed image analysis to quantify bacterial survival under fixed antibiotic concentrations (10xMIC for 4 hours) on E. coli, A. baumannii, K. pneumoniae and P. aeruginosa. This analysis was conducted using a set of antibiotics to which these strains were initially sensitive to the antibiotic employed (e.g., MIC ≤ 4 µg/ml, see also Supplementary Table 8). Survival rate was quantified by calculating the ratio of the viable cells after antibiotic treatment and the total initial viable cell count. The analysis revealed that DT permeabilizers are among the antibiotics with the best killing kinetics. In particular, SCH79797 eliminated the entire bacterial population in E. coli, A. baumannii and K. pneumoniae within 4 hours (Fig. 7a). Of note, colistin could eliminate the full bacterial population in P. aeruginosa only.”

We also calculated the resistance level (i.e., median relative MIC) reached during laboratory evolution and survival rate under each antibiotic treatment and found a significant positive correlation between the two variables (Spearman's rank correlation coefficient $\rho = 0.37$, $p = 0.0019$, see Methods).”

Minor points:

- Maharramov et al analyze cross-resistance of polymyxin (ST-permeabilizer) and DT-permeabilizer resistant lines and find that they exhibit cross-resistance to ST-permeabilizers and antibiotics for other families, respectively. In my view, the fact that ST-permeabilizers select for resistance to other ST-type antimicrobial peptides but do not do so against the other three types of antibiotics is an advantage over the type of cross-resistance selection exerted by DT-permeabilizers. Perhaps this could also be discussed throughout the manuscript.

We thank the reviewer for the insightful comment regarding the cross-resistance patterns observed in ST-permeabilizers versus DT permeabilizers. In response, we have added further discussion on this topic to highlight the advantage of DT permeabilizers in selectively avoiding cross-resistance to ST permeabilizers.

Now after line 353, we state:

“Notably, polymyxin B-adapted lines exhibited no cross-resistance to DT permeabilizer antibiotics while being resistant to ST permeabilizers (Extended Data Fig. 6). This resistance pattern highlights the potential advantage of DT-permeabilizers, as they appear less prone to promoting cross-resistance through modification of the LPS of the bacterial membrane.”

- Since the over-production of AcrAB-TolC efflux pump (able to extrude the DT-permeabilizer SCH79797) is frequently selected into the clinics, I think that SCH79797 would not be the best antibiotic to use for E. coli infections. However, it should be recommended in Klebsiella and Acinetobacter. This clarification at the species level would seem appropriate to me.

We thank the reviewer for this thoughtful comment. While we acknowledge the relevance of species-specific considerations in antibiotic recommendations, SCH79797 remains in preclinical development, and our study centers on general resistance evolution mechanisms rather than clinical recommendations. This includes considerations like drug toxicity, pharmacokinetics, and efficacy, which are beyond our current study's scope. Consequently, we avoid making specific clinical recommendations. As indicated by the title, our goal is to elucidate general principles that guide the development of resistance-free antibiotics.

- You indicate in the discussion section that “MscL remains unmutated in response to SCH79797 stress” and that “this pattern demands explanation”. I think the most reasonable explanation is given a few sentences above: “it is essential for bacterial survival”. So, I agree that it would be very interesting to look for new inhibitors of this channel, especially those that cannot be expelled by efflux pumps.

We agree, thank you.

Reviewer #2 (Remarks to the Author):

Comments to Maharramov et al. Nature Communications

This manuscript by Maharramov et al. is a very nice, interesting, important, useful and complete study of how to design antibiotics. The authors concluded that dual-target antibiotics, where one of the targets is membrane integrity, are much less prone to resistance than single-target antibiotics or even dual-target antibiotics where none of the targets is the membrane. I raised a few points I hope the authors can answer and correct.

Major points:

1) In a few places, sometimes the text gives us the (wrong) idea that there is some kind of Lamarckism (or mutation induction) in the process of antibiotic resistance. Since Luria and Delbrück (1943), we know that this is a typical process of (Darwinian) Natural Selection. Therefore, I think the authors should ameliorate a few sentences. For example, in the Abstract, line 32, the authors wrote “Third, the timeframe for resistance evolution was brief, owing to the rapid eradication of bacterial populations upon toxic antibiotic exposure.”.

As Luria and Delbrück and many others showed, resistant mutants were already resistant “before” they were in contact with the stress factor. Bacteria did not become resistant “because” they are in contact with the stress factor. That sentence refers to their beautiful experiments 7a and 7b. Yes, dual-target permeabilizers kill faster than the other antibiotic types. Still, this should be viewed as a lucky coincidence (faster killers are also the ones where resistance opportunities are rarer). It should not be considered an explanation for the inability of bacteria to “find” resistance. The paragraph in lines 342-345 (Discussion section) should also be clearer.

We modified the abstract as follows:

Lines 32-33:

“Furthermore, we detected exceptionally rapid eradication of bacterial populations upon toxic exposure to membrane targeting antibiotics.”

2) According to my previous point, I suggest that the authors comment on this coincidence.

Thank you. Now the paragraph is as follows (Lines 371-376):

“Fourth, dual-target permeabilizers exhibit strong bactericidal activity, characterized by a steep increase in efficacy within a narrow dose range compared to most other antibiotics (Fig. 7a, b). Other antibiotics, like polymyxin B and colistin, also effectively killed most bacterial populations, but simultaneously showed vulnerability to bacterial resistance.

Therefore, future studies should investigate the relationship between antibiotic's killing efficacy and resistance evolution in more detail."

3) Figures 2c, 2d, 6b and 6c are essential in this manuscript. Kruskal-Wallis analyses are insufficient because this test tells us that the groups are different, not necessarily the DT-perm antibiotics. For example, a post-hoc test would be important.

We thank the reviewer for highlighting this point, we have revised the text as follows:

- We removed the reference to Fig. 2c from our earlier statement, which previously read:

"Reassuringly, the level of resistance against DT permeabilizers was significantly lower in comparison to all other antibiotic groups (Fig. 2b, c, and Supplementary Table 2)."

We retained the reference to Fig. 2b, as it directly supports our claims.
- For Fig. 2d, we conducted additional analyses using Dunn's post-hoc test with Benjamini-Hochberg correction to account for multiple comparisons. These analyses confirmed the overall findings presented in the figure.
- In Figs. 6b and 6c, we also applied Dunn's post-hoc test with Benjamini-Hochberg correction for multiple comparisons. The results showed significant differences between DT permeabilizers and other mode-of-action groups, except for ST non-permeabilizers in Fig. 6b, and between DT permeabilizers and DT non-permeabilizers in Fig. 6c.

Now we state after line 274 that:

"We found that the number of resistance-conferring segments was significantly lower for DT permeabilizers compared to ST permeabilizers and DT non-permeabilizers (Fig. 6b, Dunn's post-hoc test with Benjamini-Hochberg correction, $p < 0.05$). Notably, when focusing exclusively on antibiotics currently in development, DT permeabilizers again showed a significantly reduced number of resistance-conferring segments compared to the others (Extended Data Fig. 7; Student's t-test, $p < 0.01$). The study of tridecaptin M152-P3 revealed an important outcome: no resistance-conferring DNA contigs were detected following selection, underscoring its potential robustness against resistance development (Fig. 6b). Additionally, SCH79797 showed the second-lowest count of DNA contigs ($n = 17$). These findings highlight the unique potential of DT permeabilizers to mitigate LPS modification-based resistance in natural environments."

Figures 6b (left) and 6c (right):

Fig. 6b. Impact of foreign DNA on antibiotic resistance. The barplot shows the impact of foreign DNA segments, derived from functional metagenomics screens, on resistance to different antibiotics. Functional selection identified 1,045 distinct antibiotic resistance-conferring DNA segments (contigs), while 4.2% of these were detected in screens against DT-permeabilizers, including 17 for SCH79797 (SCH) and 27 for POL7306 (POL). The bars represent the mean of distinct contigs that provide resistance to each group of antibiotics. Individual data points reflect specific antibiotics within these groups. Statistical analysis was performed using Dunn’s post-hoc test with Benjamini-Hochberg correction for multiple comparisons (* indicates $p < 0.05$) following Kruskal-Wallis rank sum test (chi-squared = 8.4475, df = 3, $p = 0.038$). For grouping based on modes of action, see Table 1.

Fig. 6c. The prevalence of natural *E. coli* genomes containing putative antibiotic resistance genes (ARG). The barplot illustrates the proportion of natural *E. coli* genomes that contain at least one putative ARG across the four major groups of antibiotics. The bars represent the mean of the combined fraction of *E. coli* genomes with putative ARGs among habitats that provide resistance to each group of antibiotics. Individual data points reflect mean percentages of *E. coli* genomes with putative ARGs across antibiotics per habitat. Statistical analysis was performed using Dunn’s post-hoc test with Benjamini-Hochberg correction for multiple comparisons (* indicates $p < 0.05$) following Kruskal-Wallis rank sum test (chi-squared = 9.4917, df = 3, $p = 0.023$). Notable differences were identified in the frequency of natural *E. coli* genomes that possess at least one putative ARG. For grouping based on modes of action, see Table 1.

4) I am also worried about a point in Methods (Lines 432-432) and then the interpretation of results. The authors write, “Unless otherwise indicated, cation-adjusted Mueller-Hinton Broth 2 (MHB, Millipore) medium was used throughout the study, except for SCH79797 (SCH). To maximize antibacterial activity of SCH, based on prior experience with folate biosynthesis inhibitor antibiotics, Minimal Salt...”. Could that be the reason why this DT-perm was so efficient, fast-killer, etc? Could the results of this manuscript be an artifact of the medium? This should be explained.

Thank you for raising this issue. To address the concern that the exceptional killing properties of SCH79797 might be a result of medium artifacts, we conducted additional killing experiments with SCH79797 in Mueller-Hinton Broth II (MHB) medium using *E. coli*, *A. baumannii*, and *K. pneumoniae*. Our results demonstrated that SCH79797 eradicated the entire bacterial population in all three species in the MHB medium.

Barplot illustrating the survival rates (log₁₀ -transformed) of *Acinetobacter baumannii*, *Escherichia coli*, and *Klebsiella pneumoniae* exposed 10xMIC SCH79797 across two media types (MHB and MS). Horizontal line represents the mean survival rates across biological replicates indicated by the data points. Notably, SCH79797 exposure yielded consistent bacterial eradication rates across both media types.

Minor Points:

5) Line 506: in the 96-well plates, did the authors used MHB for the DT-perm SCH?

We also used MS media for SCH79797 during this experiment. We added this information to the methods section as follows:

Lines 590-592:

*“In 96-well microtiter plates containing fresh MHB medium, **or supplemented MS medium in case of SCH79797**, ten-step serial dilutions of an antibiotic from a stock solution were made (2 wells per antibiotic concentration per strain).”*

6) I also ask the authors to elaborate more on their discussion. Why is permeabilization so important? Why not just DT-non-permeabilizers?

We write in the discussion (lines 395-404):

“These findings contrast with the wide range of functionally distinct mutations and antibiotic resistance genes observed in response to treatments with dual-target topoisomerase antibiotics. These antibiotics target two homologous intracellular protein complexes, DNA gyrase and topoisomerase IV. The corresponding evolved lines exhibited a median antibiotic resistance level that is 128 times higher than the ancestor (Fig. 2c and Supplementary Table 2). Putative resistance mutations were observed regularly in targeted proteins and major efflux pumps²⁷. In addition, functional metagenomic screens revealed that established genes involved in antibiotic target protection (e.g., QnR protein family)^{55,56} provided resistance to several dual-target topoisomerase antibiotics (Supplementary Table 7).”

We expanded it as follows:

“It is an open issue whether other non-permeabilizer dual-target antibiotics are also susceptible to resistance evolution.”

7) In addition to the previous point, I would like to ask for a small discussion about what the authors would expect with a DT-permeabilizer where both targets are the membrane.

We briefly discussed this idea as follows in the discussion (lines 413-424):

“In sum, LPS is an antibiotic target prone to resistance formation. An alternative strategy to achieve membrane permeabilization is by binding to the mechanosensitive channel MscL, as exemplified by SCH79797. MscL is a promising new target for antibacterial development, as it is essential for bacterial survival, has a highly conserved protein sequence across bacterial species, including pathogens, and is absent in mammals. MscL channels are active throughout all stages of bacterial growth, including the stationary phase, and their function does not depend on cellular metabolism or energy⁶⁷. Therefore, inhibitors targeting MscL channels should be effective against both stationary-phase cultures and dormant cells. Perhaps as a result, SCH79797 demonstrated exceptionally high antibacterial activity, successfully eliminating the entire bacterial populations within just four hours of treatment (Fig. 7 a). Finally, MscL remains unmutated in response to SCH79797 stress. This pattern demands explanation as several other diverse MscL antagonists have been discovered but their susceptibility to bacterial resistance is unknown^{18,19}.”

We feel that all discussions beyond this point would be too speculative.

8) Lines 181-185 and Figure 4a.

I made a (too) strong effort to understand these lines and figure 4a. Then I understood it, but I suggest the authors to explain better (set size, etc, etc).

We addressed this issue to improve clarity in both the text and the figure.

We have now revised the text as follows (lines 205-209):

“Surprisingly, we observed that the majority of the mutated genes (80%, n = 128) were antibiotic-specific, with none of the identified mutations shared across all five antibiotics. POL7306 exhibited a higher overlap in the set of mutated genes associated with ST permeabilizers (8.12%) compared to tridecaptin M152-P3 (2.5%) and SCH79797 (0.6%), as shown in Fig. 4a. This observation aligns with the fact that POL7306, a peptide-based antibiotic, shares structural and functional similarities with polymyxin B¹².”

To make it easier to understand, we also added the following details to the figure legend:

“Fig. 4a. Overlap of mutated genes in response to ST (polymyxin B, SPR206) and DT permeabilizers (SCH79797, tridecaptin M152-P3, POL7306). The plot shows the sets of mutated genes identified for each antibiotic and their intersections. The **vertical bars represent the number of mutated genes shared among the indicated antibiotics, as shown by the connected dots. A **single dot** indicates that the mutated genes are specific to a single antibiotic and are not shared with any others. **Horizontal bars** indicate the total number of mutated genes per antibiotic. The majority of the mutated genes are antibiotic specific (80%, $n = 128$), and none of the identified genes were shared among all five antibiotics.”**

9) Lines 211-212: in 12 strains?

We modified the sentence as follows (now lines 230-239):

*“The analysis was performed on a selected set of laboratory evolved lines ($n = 12$), all of which had developed high levels of resistance to polymyxin B. We measured the susceptibilities of these lines, along with the corresponding ancestor to ST permeabilizers (SPR206 and colistin) and DT permeabilizers (SCH79797, tridecaptin M152-P3, and POL7306). We found that adaptation to polymyxin B resulted in cross-resistance to SPR206 and colistin, while susceptibility to SCH79797 and tridecaptin M152-P3 remained unchanged (Fig. 5b and Supplementary Table 5). Among the polymyxin B resistant lines, a subset ($n = 7$) exhibited reduced susceptibility to POL7306, although these changes were less pronounced compared to those observed with SPR206 and colistin (Extended Data Fig. 6) and were specific to *K. pneumoniae* and *A. baumannii* only (Fig. 5b).*

10) Lines 275-278. The Kruskal-Wallis statistics is not testing what the sentence says (unless the authors do a post-hoc test).

Please refer to the point 3 response.

11) Lines 275-280. This is a nice point, but could it be because DT-perm. are new antibiotics? Could you comment on this in the Discussion section?

This is an important point. We emphasize that other antibiotics studied in this work include “new” antibiotics, all of which have not reached the market yet. Many such “new” antibiotics (*e.g.*, omadacycline, eravacycline, delafloxacin, gepotidacin, SPR206, sulopenem) with different modes of action were prone to resistance by acquisition of foreign DNA segments from other species. The differences between DT permeabilizers and other antibiotics hold when only “new” antibiotics were considered.

Extended Data Fig. 7. Resistance-conferring contigs in functional metagenomics screens against antibiotics currently in development. The analysis focuses on new antimicrobial compounds which have been introduced into clinical practice recently (after 2017) or are currently in development (*i.e.*, ‘recent’ antibiotics²⁸). The boxplot shows the distribution of resistance-conferring contigs identified against DT-permeabilizers and other “recent” antibiotics. Each point represents the number of contig for a given antibiotic, and boxplots show the median, first and third quartiles, with whiskers indicating the 5th and 95th percentiles. DT permeabilizers show significantly lower numbers of resistance-conferring contigs compared to other recently developed antibiotics. Statistical significance was assessed using Student's *t*-test (** indicates $p < 0.01$).

We modify the text as follows (lines 274-279):

*“We found that the number of resistance-conferring segments was significantly lower for DT permeabilizers compared to ST permeabilizers and DT non-permeabilizers (Fig. 6b, Dunn’s post-hoc test with Benjamini-Hochberg correction, $p < 0.05$). Notably, when focusing exclusively on antibiotics currently in development, DT permeabilizers again showed a significantly reduced number of resistance-conferring segments compared to the others (Extended Data Fig. 7; Student’s *t*-test, $p < 0.01$).”*

12) Lines 283-286: again the “Lamarckism” or mutation induction issue I raised in my first point. Please note that Windels et al. (ref 62) refers to “Bacterial Persistence,” as giving more opportunities to find resistance because, in the absence of antibiotics, they grow again, etc., and bacteria can mutate and become ready the next time the drug appears. It is NOT because bacteria have a wider window to “find” resistance.

We deleted these sentences, now the text is as follows (lines 313-315):

“Killing kinetics of dual-target permeabilizers

We employed image analysis to quantify bacterial survival under fixed antibiotic concentrations (10xMIC for 4 hours) on E. coli, A. baumannii, K. pneumoniae and P. aeruginosa. This analysis was conducted using a set of antibiotics to which these strains were sensitive (e.g., MIC ≤ 4 µg/mL, see also Supplementary Table 8). ...”

13) Lines 342-349: again as in point 1 and 12

We modified the text as follows (lines 371-376):

“Fourth, dual-target permeabilizers exhibit strong bactericidal activity, characterized by a steep increase in efficacy within a narrow dose range compared to most other antibiotics (Fig. 7a, b). SCH79797 was the only antibiotic that eradicated the entire bacterial population within 4 hours of treatment. Other antibiotics, like polymyxin B and colistin, also effectively killed most bacterial populations, but simultaneously showed vulnerability to bacterial resistance. Therefore, future studies should investigate the relationship between antibiotic’s killing efficacy and resistance evolution in more detail.”

14) Figure 7b: I ask the authors to mark the DT-perm drugs to make it clearer to the reader.

We modified the figure accordingly.

Fig. 7b. Dose response curves of the studied antibiotics. The figure shows bacterial survival across different concentrations of antibiotics. Survival was estimated by measuring cell viability (colony-forming units/ml) following a 4-hour exposure to different concentrations of 16 antibiotics on *E. coli* ATCC25922 (for further details, see Methods). The antibiotic concentration that kills 99.9% of the *E. coli* population is indicated as a vertical dotted line (see Supplementary Table 8). Error bars represent standard error calculated from the colony counts by Poisson's model. A Hill function was fitted to the dose-response data from two biological replicates, producing sigmoidal curves (Supplementary Table 8). The Hill coefficient reflects how steeply the survival rate decreases in response to increasing antibiotic concentration (a lower value indicates lower survival). The Hill coefficient for DT permeabilizers was significantly lower than for DT non-permeabilizers and ST non-permeabilizers (two-sample Student's t-test, $p < 0.01$), but statistically equivalent to that of ST permeabilizers (two-sample Student's t-test, $p = 0.2$). X-axis label colours indicate mode of action groups. For detailed information on antibiotic abbreviations, see Table 1.

Reviewer #3 (Remarks to the Author):

Review Maharramov et al, submitted to Nature Communications.

This is a comprehensive work on the interface between microbial evolution and antibiotic development. The manuscript is generally well written, and the vast amounts of data are nicely presented in a clear way. The data presented suggest that dual targeting can limit in vitro resistance evolution when membrane integrity is included as a target.

Points for discussion:

I have no major scientific concerns about the presented work- it is comprehensive and very impressive.

Thank you.

As a general structural comment I believe the manuscript would benefit from being even more clear with respect to whether data were obtained from the mentioned work soon to be published in Nature Microbiology- and reanalyzed here.

Throughout the current manuscript, we duly cite the Nature Microbiology manuscript and took special care to highlight where the data is coming from. For example:

Lines 103-120:

"In an earlier work, we studied the susceptibility of these antibiotics to resistance development in one multi-drug resistant and one sensitive strain each of Escherichia coli, Klebsiella pneumoniae, Acinetobacter baumannii and Pseudomonas aeruginosa²⁸. These Gram-negative species are among the critical-priority pathogens according to the World Health Organization (WHO). Using a conventional method for spontaneous resistance frequency analysis (FoR assay) at varying antibiotic concentrations, we sought to understand de novo resistance emergence. We also initiated adaptive laboratory evolution (ALE) to see how antibiotic resistance in the populations might increase over a more extended, yet fixed timeframe (roughly 120 generations). Elevated resistance was observed in ~40% (for FoR assay) and ~91% (for ALE) of the antibiotic-strain combinations we studied²⁸. We next extended our focus to include horizontally transferred resistance mechanisms. We investigated the prevalence of horizontally transferred antibiotic resistance genes in diverse resistomes, encompassing both environmental and clinical sources. Our analysis encompassed metagenomic libraries derived from three distinct sources: i) river sediment and soil samples from seven antibiotic-contaminated industrial sites near antibiotic manufacturing plants in India, representing the anthropogenic soil microbiome; ii) faecal samples from ten Europeans with no antibiotic consumption for at least one year before collection, representing the gut microbiome; and iii) a composite

sample from 68 multi-drug resistant bacteria, either isolated in medical settings or acquired from bacterial 120 strain collections, representing the clinical microbiome. "

Lines 120-122:

"Utilizing established functional metagenomic techniques, we identified 690 unique DNA segments capable of significantly enhancing resistance to one of each antibiotic²⁸."

Lines 161-163:

"Using the same set of bacterial strains, we initiated adaptive laboratory evolution with the aim to maximize the level of antibiotic resistance in the populations achieved during a longer, but fixed time period²⁸."

Lines 269-271:

"Using advanced functional metagenomic techniques, we earlier found 551 DNA segments that boosted resistance to the antibiotics tested here²⁸."

Additionally, we included the reference for the Daruka *et al.*, paper to the following sentence:

Lines 138-140:

"To explore first-step resistance, we previously used a standard protocol for spontaneous frequency-of-resistance analysis (FoR assay) at multiple concentrations of each antibiotic^{10,28,31-33}."

With the current version of ref 28 only present at bioRxiv I really would not know (in full awareness that the final published version could differ). To that end, another suggestion would be to bring the other paper up in the discussion to highlight contrasting points.

The two papers are complementary in scope and findings. The previous paper primarily aimed to provide a comprehensive framework for predicting the potential for resistance evolution against new antibiotic candidates, comparing them to in-use antibiotics. In contrast, the current manuscript focuses specifically on dual-target permeabilizer antibiotics and explores whether this approach can limit resistance, particularly in Gram-negative pathogens. Our present study builds on some of the data from the earlier research, reanalyzing it with a distinct focus on antibiotics that simultaneously impair membrane integrity while targeting a second cellular function. This reanalysis did not involve modification of the original data but allowed us to test more specific hypotheses about resistance mechanisms and their evolution under different selective pressures.

How certain can the authors be about the distinct classes of non-permeabilizers? Recent data suggest that classical non-permeabilizing drugs may interact with the membrane (<https://doi.org/10.1016/j.bbamem.2020.183448>).

Our classification was based on our own experiment, where we measured the ability of the different antibiotics to permeabilize the bacterial membrane (Extended Data Fig. 1). Thus, we are convinced that even if non-permeabilizer antibiotics can interact with the membrane (as suggested by the cited paper) these will not initiate the permeabilization of it.

FoR analyses: I could not find the actual frequencies anywhere- I also checked in the bioRxiv manuscript (Daruka et al)- presenting the frequency data of mutations at different concentrations would be interesting to see

The ultimate goal with this assay was different. We aimed to demonstrate the highest level of resistance that can be achieved in a short period of time. As noted earlier, the frequency of resistance is not a reliable metric to predict resistance evolution in clinical settings: Morten O. A. Sommer et al. 2017, Nature Reviews Microbiology, DOI: [10.1038/nrmicro.2017.75](https://doi.org/10.1038/nrmicro.2017.75), Figure 3. "The in vitro rate of resistance mutations does not correlate with the burden of resistance in the clinic." Thus we decided to use the fold change in MIC rather than the frequency of resistance metric.

Line 188: “..to Dt permeabilizers..” right?

We intended to refer to adaptation to "ST permeabilizers," specifically including SPR206 and polymyxin B, along with POL7306 and tridecaptin-M152-P3 (Fig. 4b).

To improve clarity, we have revised the text to explicitly list the names of these antibiotics as follows (lines 211-212):

“Genes involved in cell envelope biogenesis and regulation were commonly mutated in lines adapted to SPR206, polymyxin B, POL7306, and tridecaptin-M152-P3 (Fig. 4b).”

Mobile resistance (from line 242)- these experiments were limited by 5 kb inserts- would it not be worth including this limitation explicitly? Vancomycin resistance for example is encoded by an 10,8 kb transposon.

Yes, this could be a potential limitation. However, as shown previously, the length distributions of the known antibiotic-resistance genes are generally well within this fragment size range (see PMID: 30559406). Please also note that vancomycin is Gram-positive specific antibiotic, while we focus on Gram-negative pathogens.

We now mention this limitation on the Methods section as follows (lines 673-675):

“The length distributions of known antibiotic-resistance genes generally fall within this fragment size range, however some larger resistance elements may not be captured.”

Include materials and methods in main text and not separate Supplementary Information

We moved the requested sections to the main Methods section.

Fig. 2d. boxes missing for Pa and Ab

We have now clarified in both the manuscript and the figure legend that the missing data for *Pseudomonas aeruginosa* (PA) and *Acinetobacter baumannii* (AB) are due to decreased susceptibility (e.g.: MIC > 4 µg/mL), which excluded these antibiotic-strain combinations from laboratory evolution experiments.

Line 349: include PK/PD considerations as well as more in vivo like conditions to fully evaluate resistance potential?

In the discussion we write (line 426-429):

*“Our study investigated the effects of diverse antibiotics on bacterial resistance evolution to elucidate factors contributing to antibiotic efficacy. We identified key principles of antibiotic action that minimize resistance and proposed potential directions for future antibiotic development. However, further *in vivo* experiments are needed to enhance and validate the findings presented here.”*

Reviewer #4 (Remarks to the Author):
